



# Solid-state $^{13}$C-NMR spectroscopic determination of sidechain mobilities in zirconium-based metal-organic frameworks

Günter Hempel[1], Ricardo Kurz[1], Silvia Paasch[2], Kay Saalwächter[1], and Eike Brunner[2]

[1]Martin-Luther-Universität Halle-Wittenberg, Institut für Physik – NMR, Betty-Heimann-Str. 7, 06120 Halle, Germany
[2]Technische Universität Dresden, Fakultät für Chemie und Lebensmittelchemie, Bioanalytische Chemie, 01062 Dresden, Germany

**Correspondence:** Günter Hempel (guenter.hempel@physik.uni-halle.de)

**Abstract.** Porous Interpenetrated Zirconium-Organic Frameworks (PIZOFs) are a class of Zr-based metal-organic frameworks (MOFs) which are composed of long, rodlike dicarboxylate linkers and $Zr_6O_4(OH)_4(O_2C)_{12}$ nodes. Long oligoethylene glycol or aliphatic side chains are covalently attached to the linker molecules in the case of PIZOF-10 and PIZOF-11, respectively. These side chains are supposedly highly mobile thus mimicking a solvent environment. It is anticipated that such MOFs

could be used as solid catalyst - the MOF - with pore systems showing properties similar like a liquid reaction medium. To quantify the side chain mobility, we have here applied different 1D and 2D NMR solid-state spectroscopic techniques like cross polarization (CP) and dipolar-coupling chemical-shift correlation (DIPSHIFT) studies. The rather high $^1$H-$^{13}$C CP efficiency observed for the $CH_2$ groups of the side chains indicates that the long side chains are unexpectedly immobile or at least that their motions are strongly anisotropic. More detailed information about the mobility of the side chains was then obtained from

DIPSHIFT experiments. Analytical expressions for elaborate data analysis are derived. These expressions are used to correlate order parameters and slow motional rates with signals in indirect spectral dimension thus enabling the quantification of order parameters for the $CH_2$ groups. The ends of the chains are rather mobile whereas the carbon atoms close to the linker are stronger spatiallly restricted in mobility.

## 1   Introduction

Metal-organic frameworks (MOFs) are crystalline porous materials (Kitagawa et al., 2004; Ferey, 2008; Tranchemontagne et al., 2009; Kaskel, 2016) composed of organic and inorganic building units forming networks with micro- and/or mesopores. Due to promising properties such as an extraordinarily high specific surface area and gas storage capacity, MOFs are assumed to find numerous applications, e.g., in gas storage and separation, drug delivery, sensing, and wastewater treatment (Horcajada et al., 2006; Ke et al., 2011; Maurin et al., 2017; Li et al., 2019; Lawson et al., 2021; Maranescu and Visa, 2022; Liu et al.,

2023). Favorable MOFs for catalysis (Pascanu et al., 2019; Remya and Kurian, 2019; Bavykina et al., 2020; Gao et al., 2022) should be stable at elevated temperatures and in the presence of moisture. Most members of the PIZOF (Porous Interpenetrated Zirconium-Organic Framework) family (Schaate et al., 2011; Roy et al., 2012; Lippke et al., 2017) are very stable and insensitive to moisture in contrast to other MOFs containing, e.g., Zn instead of Zr (Kaye et al., 2007; Feng et al., 2010) which makes this type of MOFs particularly interesting for catalytic applications. PIZOFs contain long dicarboxylates as linkers (cf. Fig. 1)



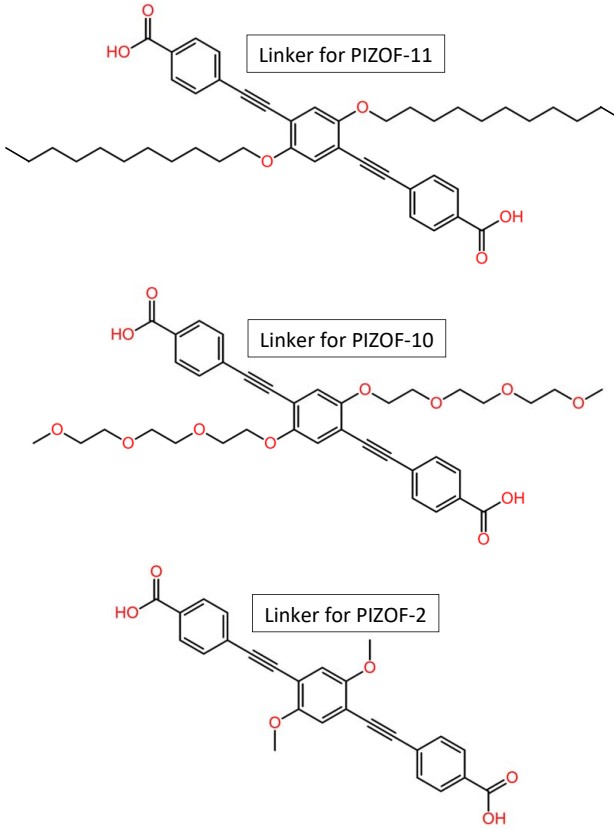

**Figure 1.** Structural formulae of the linkers for the herein studied PIZOF's PIZOF-2, PIZOF-10, and PIZOF-11

and $Zr_6O_4(OH)_4(O_2C)_{12}$ as nodes. The surface properties of the internal pore system including its polarity can be tuned in PIZOFs by adding appropriate side chains to the linkers. Examples are PIZOF-10 and PIZOF-11 (see Fig. 1). Both exhibit high thermal stability and resistance against atmospheric moisture like other members of the PIZOF family (Lippke et al., 2017). The aliphatic side chains make the internal surface of PIZOF-11 hydrophobic whereas PIZOF-10 with its oligoethylene glycol side chains is rather hydrophilic.

The long oligoethylene glycol or aliphatic side chains in PIZOF-10 and PIZOF-11 are supposed to be highly flexible thus providing a liquid-like environment in the remaining pores. In order to test this hypothesis, solid-state NMR techniques were applied to evaluate the side chain mobility. Solid-state NMR spectroscopy is a powerful tool for determining structural parameters and detecting mobile structural elements in biological systems as well as various materials (Renault et al., 2010; Paasch and Brunner, 2010; Duer, 2002). The framework structure of MOFs can be studied in general by $^1$H, $^{13}$C or $^{17}$O magnetic angle

spinning (MAS) NMR spectroscopy (Loiseau et al., 2005; Klein et al., 2012; Lucier et al., 2018; Bignami et al., 2018; Brunner and Rauche, 2020). Furthermore, nuclei such as $^{27}$Al (Jiang et al., 2010; Lieder et al., 2010; Petrov et al., 2020), $^{71}$Ga (Volkringer et al., 2007; Hajjar et al., 2011), $^{43}$Sc (Mowat et al., 2011), and others (He et al., 2014) can be used in order to





detect the environment of the central metal atom. NMR spectroscopy also allows the study of host/ guest interactions with adsorbed species (Wong et al., 2019; Witherspoon et al., 2018; Yan et al., 2017; Wittmann et al., 2019).

It is well-known that the efficiency of the cross polarization (CP) experiment (Pines et al., 1973; Metz et al., 1994; Fung et al., 2000) is strongly influenced by the presence of thermal motions (Schulze et al., 1990). This has been exploited previously in order to qualitatively characterize the mobility of aliphatic chains, e.g. of n-alkylsilanes bonded to silica surfaces (Sindorf and Maciel, 1983), of surfactants in mesoporous MCM-41 materials (Simonutti et al., 2001), and alkanes grafted to silica for chromatography (Pursch et al., 1996).

The motion of the side chains is expected to be anisotropic because of its fixation at the middle ring of the linker backbone. The degree of anisotropy (expressed e.g. by the order parameter S) resolved for individual atomic positions might be an important characteristic of the chain motion. More detailed information can be obtained by some 2D NMR methods. In the work presented here, the DIPSHIFT (dipolar-coupling chemical-shift correlation) technique was applied in order to site-specifically determine order parameters. It was developed as separated-local-field experiment under MAS (Munowitz et al.,

1981; Schaefer et al., 1983). Applications made use of the possibility to determine the residual dipolar coupling site-specifically (de Azevedo et al., 2008; Bärenwald et al., 2016) and a recent paper compares the results with those of rotational echo double resonance (REDOR, Jain et al. (2019)). The DIPSHIFT experiment enables correlation of isotropic chemical shifts, i.e. atomic positions, in direct dimension ($t_2$) with signals in the indirect dimension ($t_1$) corresponding to the dipolar on-resonance $^{13}$C free induction decay (FID) under MAS between subsequent rotational echoes. The latter is true under the condition that protons coupled to the considered nucleus do not undergo mutual interactions and provided that Fourier transform is performed in direct

dimension only. The depth of the minimum between consecutive echoes is a measure for the (residual) dipolar coupling which itself contains information about the order parameter. The decrease of the 2nd echo with respect to the first one represents the influence of intermediate motions (intermediate-motional $T_2$ effect).

To determine the mobility site-selectively, $^{13}$C signal assignment was necessary. This could be achieved by the application of

techniques such as solid-state attached proton test (APT, Lesage et al. (1998b); Hoffmann et al. (2012)), $^1$H-$^{13}$C heteronuclear correlation (HETCOR), and $^1$H-$^{13}$C heteronuclear multiple-quantum coherence (HMQC, Lesage et al. (1998a)).

The analysis of DIPSHIFT data was in the past usually performed by comparison with numerically calculated data under parameter variation. In this paper, analytical expressions were derived and used. This allowed to estimate the parameter set for best data fitting in a more efficient way. Intermediate motions were included in the analytical expressions by application of the

Anderson-Weiss procedure. Specifically, we focus on the conditions under which the signal damping arising from intermediate motions can be described by an exponential function. The theoretical background of this procedure is described in the next section.





## 2 Equations / models for DIPSHIFT data evaluation

### 2.1 Derivation

MAS experiments on samples with anisotropic spin interactions result in spectra with spinning sidebands (SSB) and in FID's containing echo trains. Provided that the anisotropic interaction is described by a second-order tensor, the SSB intensities can be efficiently calculated by polynomial expressions following Hempel et al. (2021). Eqn. (21) in that paper gives the general equation for the intensity $S_m(D/\omega_{\mathrm{r}}, \eta)$ of the $m$-th order sideband; the parameters in the argument list are the anisotropy $D$ of the tensor, the angular spinning frequency $\omega_{\mathrm{r}}$, and the asymmetry parameter $\eta$ of the tensor. A closed-form evaluation of

this equation is hardly possible; equations for SSB which are ready for use must be obtained by symbolic-language software. Examples for low-order SSB are given in the mentioned paper.

The FID can be calculated as Fourier synthesis if the SSB intensities are known:

$$F(t) = \sum_{m=-\infty}^{\infty} S_m \left( \frac{D}{\omega_{\mathrm{r}}}, \eta \right) e^{im\omega_{\mathrm{r}}t} \tag{1}$$

Applying this formalism to spin systems where chemical-shift anisotropy is the dominating interaction is straightforward. In

the case of dipolar interaction, however, some special aspects have to be considered:

- Fast anisotropic thermal motion: The tensor is subjected to a partial averaging process over all possible states which occur during this motion. Then anisotropy and asymmetry parameter of this averaged tensor have to be inserted into the SSB polynomials. In this paper, this case has to be considered for modelling two-site jumps.

- Dipolar interaction with more than one spin: Tensors of the individual couplings have to be added up. If, however, the

neighbouring spins interact among themselves, it is possible that the total interaction is represented by a tensor of order > 2. In the latter case the polynomial formulae mentioned above would not be applicable.

- In the case of multispin interactions, different combinations of spin orientations have to be taken into account; they can lead to different effective tensors. Each of these tensors corresponds to a particular set of SSB $\left\{ S_m^k \right\}$ where $k$ indicates the respective combination of spin orientations. The SSB intensity which appears in the experiment is the average of

the SSB intensities belonging to all combinations: $S_m = \sum_k S_m^k$. This procedure is applied below for the calculation of FID's of both $CH_2$ and $CH_3$ groups.

### 2.2 Transformation to a cosine series

In this paper we consider dipolar interactions between spins 1/2. The observed spin can be oriented parallel or antiparallel to $\mathbf{B}_0$ with approximately equal probabilities at not too low temperatures. A change of orientation of the observed spin inverts

the anisotropy of the dipolar tensor assuming the configuration of the coupled spins remains unchanged. We thus always have pairs of tensors with inverted anisotropy ($D$ and ($-D$)) but equal asymmetry parameter. Applying the relation (see Hempel





et al. (2021))

$$S_m \left( \frac{D}{\omega_\mathrm{r}}, \eta \right) = S_{-m} \left( -\frac{D}{\omega_\mathrm{r}}, \eta \right) \tag{2}$$

the exponential series of eqn. (1) can be transformed always into a cosine series:

$$F(t) = \sum_{m=0}^{\infty} C_m \cos m\omega_\mathrm{r} t \tag{3}$$

where

$$C_0 := S_0 ; \quad C_m \left( \frac{D}{\omega_\mathrm{r}}, \eta \right) := S_m \left( \frac{D}{\omega_\mathrm{r}}, \eta \right) + S_{-m} \left( \frac{D}{\omega_\mathrm{r}}, \eta \right) = S_m \left( \frac{D}{\omega_\mathrm{r}}, \eta \right) + S_m \left( -\frac{D}{\omega_\mathrm{r}}, \eta \right) \tag{4}$$

Therefore, the $C_m$ contain only even powers of $D/\omega_\mathrm{r}$.

In the following, the polynomial equations are established for four situations which may occur in the PIZOF samples.

## 2.3 Model 1: Ensemble of isolated spin pairs

The coefficients are denoted by $C_m^\mathrm{IS}$ and can be taken immediately from the generic equation by using the dipolar coupling constant $D_0$ as anisotropy and zero asymmetry parameter. As an example, up to 12th order we have:

$$C_0^\mathrm{IS} = 1 - \frac{3}{20} \left( \frac{D_0}{\omega_\mathrm{r}} \right)^2 + \frac{227}{181\,440} \left( \frac{D_0}{\omega_\mathrm{r}} \right)^4 - \frac{1103}{2\,306\,304} \left( \frac{D_0}{\omega_\mathrm{r}} \right)^6 + \frac{22\,859}{1\,792\,327\,680} \left( \frac{D_0}{\omega_\mathrm{r}} \right)^8$$
$$- \frac{3\,308\,407}{14\,302\,774\,886\,400} \left( \frac{D_0}{\omega_\mathrm{r}} \right)^{10} + \frac{108\,665\,671}{35\,886\,962\,442\,240\,000} \left( \frac{D_0}{\omega_\mathrm{r}} \right)^{12} \tag{5}$$

The polynomial equations for $m = 1$ to 3 are given in the appendix.

## 2.4 Model 2: Spin pairs with fast 2-site jumps

„Fast" means that the motional average is already complete at times which are relevant for the experiment. The coefficients for model 2 are denoted by $C_m^\mathrm{IS\text{-}j}$. The characteristical tensor is the average $\langle \mathbf{D} \rangle$ of the tensors of both sites $\mathbf{D}_1$ and $\mathbf{D}_2$, see Fig. 2. We suppose that the sites are occupied with equal probability. The angle between the dipolar axes of both sites is $2\alpha$. In a frame where the $z$ axis is along the bisector, the $y$ axis is perpendicular to the plane spanned by the IS connection vectors of both sites, and the $x$ axis is in this plane perpendicular to the bisector, both tensors are represented by the matrices $D_1$ and $D_2$ as

$$D_{1;2} = D_0 \begin{pmatrix} \frac{1}{2} - P_2(\cos\alpha) & 0 & \pm\frac{3}{4}\sin 2\alpha \\ 0 & -\frac{1}{2} & 0 \\ \pm\frac{3}{4}\sin 2\alpha & 0 & P_2(\cos\alpha) \end{pmatrix} \tag{6}$$

The average of both matrices is diagonal in this bisector frame:

$$\langle D \rangle = D_0 \begin{pmatrix} \frac{1}{2} - P_2(\cos\alpha) & 0 & 0 \\ 0 & -\frac{1}{2} & 0 \\ 0 & 0 & P_2(\cos\alpha) \end{pmatrix} \tag{7}$$



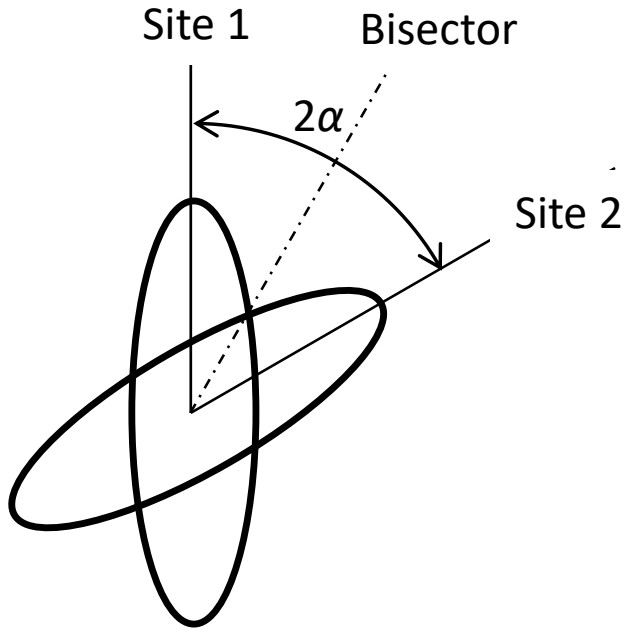

**Figure 2.** Two-site jumps between positions differing by an angle of $2\alpha$.

Particularly, in this work, 180° ring flips are of interest. That means the flip angle for the CH bonds is $2\alpha = 120°$:

$$\langle D \rangle_{120°} = D_0 \begin{pmatrix} \frac{5}{8} & 0 & 0 \\ 0 & -\frac{1}{2} & 0 \\ 0 & 0 & -\frac{1}{8} \end{pmatrix} \tag{8}$$

In the latter case, the anisotropy is $\frac{5}{8} D_0$ and the magnitude of the asymmetry parameter is $|\eta| = \frac{3}{5}$. The cos Fourier coefficients are

$$C_m^{\text{IS-j}} = C_m \left( \frac{5 D_0}{8 \omega_r} , \frac{3}{5} \right) \tag{9}$$

The explicit polynomial equations can be found in the appendix.

### 2.5  Model 3: IS$_2$ spin system

Here it is assumed that the observed spin $I$ interacts with two neighbouring spins $S$. The coupling constant for each single coupling is equal to $D_0$, i.e. the distances between $I$ and both $S$ are assumed to be equal. There are four combinations of the $S$ spin orientations. The two combinations where the two $S_z$ are parallel generate the tensor $\mathbf{D}_\Sigma$ and the two ones with antiparallel $S_z$ generate the tensor $\mathbf{D}_\Delta$. In the bisector frame (see Fig. 3), the tensor matrices are the same as in eqn. (6). The





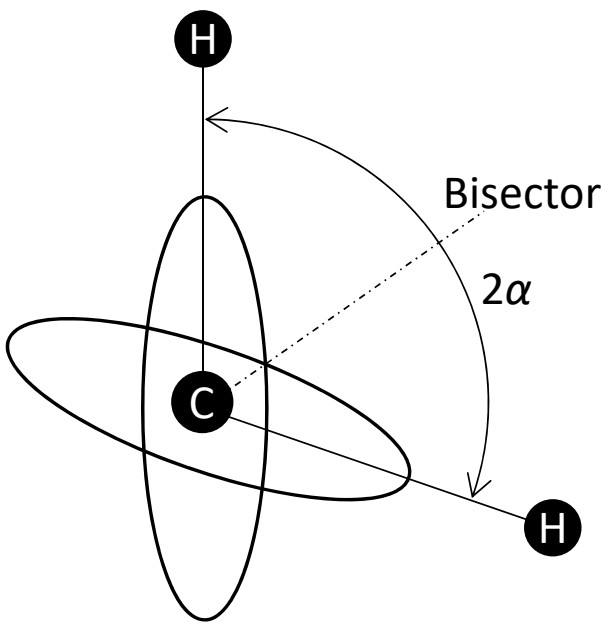

**Figure 3.** $^1$H positions the angle between the connection vectors is $2\alpha$. Supposing tetrahedral symmetry, $\alpha = \arccos\left(1/\sqrt{3}\right)$

matrices of sum and difference tensor for tetrahedral symmetry are

$$
D_\Sigma = D_0 \begin{pmatrix} 1 & 0 & 0 \\ 0 & -1 & 0 \\ 0 & 0 & 0 \end{pmatrix}, \qquad D_\Delta = D_0 \begin{pmatrix} 0 & 0 & \sqrt{2} \\ 0 & 0 & 0 \\ \sqrt{2} & 0 & 0 \end{pmatrix}, \tag{10}
$$

respectively. From $D_\Sigma$ we can read off directly the anisotropy as $D_0$ and the asymmetry parameter $\eta = 1$. $D_\Delta$ has an eigenvalue vector $\left(0, D_0\sqrt{2}, -D_0\sqrt{2}\right)$ which gives anisotropy $D_0\sqrt{2}$ and $\eta = 1$. Hence the cosine coefficients for a tetrahedral IS$_2$ spin
system are

$$
C_m^{\text{IS2}} = \frac{1}{2}\left[ C_m\left(\frac{D_0}{\omega_r}, 1\right) + C_m\left(\frac{D_0\sqrt{2}}{\omega_r}, 1\right) \right] \tag{11}
$$

The polynomial equations can again be found in the appendix.

### 2.6 Model 4: Rapidly rotating methyl groups

Here we consider the special case of an IS$_3$ spin system: The observed spin I resides in the middle of a tetrahedron, the three IS
140 bonds have equal length and point into three corners of this tetrahedron, and the rotation axis points into the remaining corner,



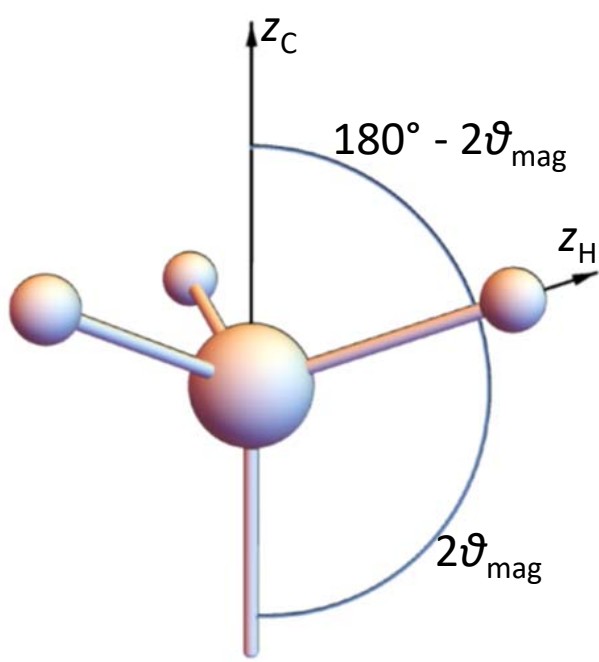

**Figure 4.** Geometry of a CH$_3$ group.

see Fig. 4. The tensor of a single CH coupling in a frame, the $z$ axis of which is along the rotation axis, has the matrix

$$D_1 = D_0 \begin{pmatrix} \frac{1}{6} + \frac{2}{3}\cos 2\varphi & \frac{2}{3}\sin 2\varphi & \frac{\sqrt{2}}{3}\cos\varphi \\ \frac{2}{3}\sin 2\varphi & \frac{1}{6} - \frac{2}{3}\cos 2\varphi & \frac{2}{3}\sin 2\varphi \\ \frac{\sqrt{2}}{3}\cos\varphi & \frac{2}{3}\sin 2\varphi & -\frac{1}{3} \end{pmatrix} ; \tag{12}$$

where $\varphi$ is the instantaneous rotation angle around the $z$ axis. Averaging over this angle, which is equivalent to fast-limit time averaging corresponding to a methyl group at not too low temperatures, gives for all three couplings a tensor matrix which is

diagonal in this frame:

$$\langle D \rangle_\varphi = D_0 \begin{pmatrix} 1/6 & 0 & 0 \\ 0 & 1/6 & 0 \\ 0 & 0 & -1/3 \end{pmatrix} \tag{13}$$

There are 8 spin orientation permutations of the three $S$ spins, see table 2.6:



| Spins | Total tensor | sum matrix | anisotropy | $\eta$ |
|---|---|---|---|---|
| ↑↑↑ | $\mathbf{D}_1 + \mathbf{D}_2 + \mathbf{D}_3$ | $3\langle D \rangle_\varphi$ | $-D_0$ | 0 |
| ↓↑↑ | $-\mathbf{D}_1 + \mathbf{D}_2 + \mathbf{D}_3$ | $\langle D \rangle_\varphi$ | $-D_0/3$ | 0 |
| ↑↓↑ | $\mathbf{D}_1 - \mathbf{D}_2 + \mathbf{D}_3$ | $\langle D \rangle_\varphi$ | $-D_0/3$ | 0 |
| ↑↑↓ | $\mathbf{D}_1 + \mathbf{D}_2 - \mathbf{D}_3$ | $\langle D \rangle_\varphi$ | $-D_0/3$ | 0 |
| ↑↓↓ | $\mathbf{D}_1 - \mathbf{D}_2 - \mathbf{D}_3$ | $-\langle D \rangle_\varphi$ | $D_0/3$ | 0 |
| ↓↑↓ | $-\mathbf{D}_1 + \mathbf{D}_2 - \mathbf{D}_3$ | $-\langle D \rangle_\varphi$ | $D_0/3$ | 0 |
| ↓↓↑ | $-\mathbf{D}_1 - \mathbf{D}_2 + \mathbf{D}_3$ | $-\langle D \rangle_\varphi$ | $D_0/3$ | 0 |
| ↓↓↓ | $-\mathbf{D}_1 - \mathbf{D}_2 - \mathbf{D}_3$ | $-3\langle D \rangle_\varphi$ | $D_0$ | 0 |

Regarding that the anisotropy enters the $C_m$ equations in even powers only, the cosine coefficients are

$$C_m^{\text{CH3}} = \frac{1}{4} C_m \left( \frac{D_0}{\omega_{\text{r}}} , 0 \right) + \frac{3}{4} C_m \left( \frac{D_0}{3\omega_{\text{r}}} , 0 \right) \tag{14}$$

The explicit polynomial equations can again be found in the appendix.

### 2.7 Influence of remote protons

Because of the $r^{-3}$ dependence of the dipolar coupling, the dipolar environment of protonated carbons is essentially determined by the directly bound proton(s) located at a distance of about 1.1 Å. The situation is less unique for nonprotonated carbons, where often several remote protons with similar distances to the regarded carbon are relevant. Then, many details of the molecular geometry have to be considered to establish a model function. On the other hand, a restriction to allegedly dominant protons can be a reason for incorrect data analysis. As an attempt, the cosine coefficients $C_m$ were calculated for the triple-bonded carbons considering the closest protons of each neighboring ring. Comparing this model function with experimental data, however, showed that the former does not reflect the real situation; the residual couplings obtained by the analysis are still larger that the full coupling expected from the model. Consequently we restricted the DIPSHIFT evaluations to protonated carbons.

### 2.8 Fast anisotropic motion

As mentioned above, fast-limit averaging means that motional averaging is complete at times which are relevant for the experiment. The effect of fast-motional averaging can be described by tensor matrices with elements changed compared to the rigid case, hence also anisotropy and asymmetry parameter change. Two general cases can be distinguished:

– The eigenvalue spectrum is reduced in such way that all three eigenvalues are decreased by a common factor which is called order parameter $S$. Then also the anisotropy is reduced by the same factor; the asymmetry parameter stays constant. The variable $D_0$ in the functions from above is then replaced by the residual dipolar coupling $D_{\text{res}} = S \cdot D_0$. Relevant examples are uniaxial rotation or rotational jumps between 3 or more equally distributed and populated sites.





– The tensor eigenvalues change during the averaging process in different manner. Then the introduction of a single order parameter will not provide an adequate description of the situation, anisotropy and asymmetry parameter change non-trivially and a new function has to be used. As an example, consider the above case of 2-site jumps: The function $C_m^{\mathrm{IS}}$ had to be replaced by $C_m^{\mathrm{IS\text{-}j}}$.

## 2.9 Intermediate motions

Intermediate motions, i.e. motions with a rate in the range of the static-limit dipolar coupling constant and/or the spinning frequency, interfere with the refocusing effect of MAS. This manifests itself by a damping of the rotational echoes or even by their complete disappearance. A rather weak echo damping is observed in our data sets. To analyze these we need an analytical expression for this case, for which we took a heuristic approach of multiplying of the rigid-lattice model functions derived above by an exponential damping function. The relevant questions are under which conditions this can be an appropriate approximation and whether the fitted decay constant contains information on the motional timescale.

An approximate analytical expression for the MAS FID $F_{\mathrm{AW}}(t)$ during slow or intermediate motion was derived by Hirschinger (2006) using the Anderson-Weiss (AW) formalism (Anderson and Weiss, 1953):

$$F_{\mathrm{AW}}(t) = \exp\left\{ -\int\limits_0^t (t-\tau)\, K(\tau)\, \mathrm{d}\tau \right\} \tag{15}$$

$K(\tau)$ is the orientation autocorrelation function of the motion under consideration. In the case of MAS, the expression derived by Clough and Gray (1962)

$$K(\tau) = e^{-\tau/\tau_{\mathrm{c}}} \left( \frac{2}{3}\cos\omega_{\mathrm{r}}\tau + \frac{1}{3}\sin\omega_{\mathrm{r}}\tau \right) , \tag{16}$$

containing an assumed exponential decay describing the random motion with correlation time $\tau_{\mathrm{c}}$, multiplied by the coherent (oscillatory) correlation function of the MAS rotation, was inserted into eqn. (15). This gave

$$
\begin{aligned}
F_{\mathrm{AW}}(t) = \exp\Bigg\{ &-\frac{1}{3}M_2 t \left( \frac{2\tau_{\mathrm{c}}}{1+(\omega_{\mathrm{r}}\tau_{\mathrm{c}})^2} + \frac{\tau_{\mathrm{c}}}{1+4(\omega_{\mathrm{r}}\tau_{\mathrm{c}})^2} \right) \\
&+ \frac{1}{3}M_2\tau_{\mathrm{c}}^2 \left[ 2\frac{1-(\omega_{\mathrm{r}}\tau_{\mathrm{c}})^2}{\left(1+(\omega_{\mathrm{r}}\tau_{\mathrm{c}})^2\right)^2}\left(1-e^{-t/\tau_{\mathrm{c}}}\cos\omega_{\mathrm{r}}t\right) + \frac{1-4(\omega_{\mathrm{r}}\tau_{\mathrm{c}})^2}{\left(1+4(\omega_{\mathrm{r}}\tau_{\mathrm{c}})^2\right)^2}\left(1-e^{-t/\tau_{\mathrm{c}}}\cos 2\omega_{\mathrm{r}}t\right) \right] \\
&+ \frac{4}{3}M_2\tau_{\mathrm{c}}^2 e^{-t/\tau_{\mathrm{c}}} \left[ \frac{\sin\omega_{\mathrm{r}}t}{\left(1+(\omega_{\mathrm{r}}\tau_{\mathrm{c}})^2\right)^2} + \frac{\sin 2\omega_{\mathrm{r}}t}{\left(1+4(\omega_{\mathrm{r}}\tau_{\mathrm{c}})^2\right)^2} \right] \Bigg\}
\end{aligned}
\tag{17}
$$

This equation describes a dipolar FID, which is an echo train for large $\tau_{\mathrm{c}}$ or a monotoneous decay for fast motion, i.e. short $\tau_{\mathrm{c}}$. The echo-train structure is widely lost already in the vicinity of $\omega_{\mathrm{r}}\tau_{\mathrm{c}} = 1$. However, it is an approximate expression because the AW approach restricts the cumulant expansion of the distribution of the reduced phase $\varphi := (1/t)\int_0^t \omega(t_1)\,\mathrm{d}t_1$ to the second-order term. Hence, all reduced-phase distributions are regarded as Gaussian distributions within the AW treatment. This could

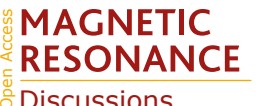

be a good approximation for short times ($t \ll M_2^{-1/2}$) (de Azevedo et al., 2008) as well as for long times ($t \gg \tau_c$). For short times, higher cumulants, being the larger the stronger the spectrum deviates from Gaussian shape (order $2n$), are still without influence because they are connected with $t^{2n}/(2n)!$ ; for long times, the reduced-phase distribution changes gradually toward a Gaussian distribution. This can be rationalized by means of the central limiting theorem, which states that the cumulants of order >2 decrease if a stochastic variable (here: reduced phase) is the sum or integral of another stochastic variable (here:

frequency).

In between we expect a "gap of validity". Obviously, for sufficiently short correlation times ($\tau_c \ll M_2^{-1/2}$) there is an overlap of both regions; here a gap of validity does not appear. For a rigid lattice, however, this gap ranges formally until infinity. For DIPSHIFT applications we have to consider that the data are usually plotted versus $t_1/T_r = \omega_r t_1/(2\pi)$ in the interval [0,1]. To ensure that this interval does not reach essentially into the validity gap, $\max t_1 = T_r$ should be smaller than $\sqrt{5}/D_0$, in other

words $D_0/\omega_r <\approx 0.5$. In fact, a numerical comparison of model function 1 with eqn. (17) for $\tau_c \to \infty$ and $M_2 \to D_0^2/5$ shows that deviations become relevant if this threshold is exceeded.

de Azevedo et al. (2008) investigated the quality of the AW approximation for the DIPSHIFT experiment by comparison of eqn. (17) with numerically calculated data. Their results show that the deviations between eqn. (17) and experimental data are maximum in the intermediate region. Cobo et al. (2014) did that in a similar manner for a related (recoupled) experiment,

rec-DIPSHIFT. This concerns situations where the echo structure of the FID is damped strongly.

Conversely, if we observe a weak damping of the echo train we may conclude that the influence of intermediate motions is negligible (except fast motions, see above). This means that in this case, $\omega_r \tau_c \gg 1$. With this condition, eqn. (17) can be simplified to

$$F_{\text{AW}}(t) \approx F_{\text{rigid}}(t) \cdot \exp\left\{-\frac{t}{T_{2\text{MAS}}}\right\} = F_{\text{rigid}}(t) \cdot \exp\left\{-\frac{1}{f_r T_{2\text{MAS}}} \cdot \frac{t}{T_r}\right\} \ , \tag{18}$$

where $F_{\text{rigid}}(t)$ denotes the FID without slow thermal motion; the damping constant is defined as

$$\frac{1}{T_{2\text{MAS}}} := \frac{3}{4}\frac{M_2}{\omega_r^2 \tau_c} \ . \tag{19}$$

It is notable that the damping constant is proportional to $f_r^{-2}$ in a plot vs. $t$ and even proportional to $f_r^{-3}$ in the DIPSHIFT-typical plot vs. $t/T_r$. Numerical studies showed that the condition $\tau_c \ll T_r$ is tied to the intensity of the first rotational echo being at least half of the initial value.

The static part of eqn. (17) is obtained for $\tau_c \to \infty$ as

$$F_{\text{rigid,AW}} = \exp\left\{-\frac{M_2}{3\omega_r^2}\left[2(1-\cos\omega_r t) + \frac{1}{4}(1-\cos 2\omega_r t)\right]\right\} \tag{20}$$

We have confirmed numerically that relevant deviations from the polynomial formulae occur if $D_0/\omega_r$ is comparable to or larger than 0.5. Consequently, $F_{\text{rigid}}(t)$ in eqn. (18) is represented by eqn. (3), where the $C_m$ are the corresponding coefficients derived for the four models shown above.





## 2.10 Second moment

The equations derived by the AW approach contain the second moment $M_2$ of the overall dipolar interactions:

$$M_2 \equiv \langle \omega^2 \rangle = \int \omega^2 \, S(\omega) \mathrm{d}\omega \tag{21}$$

$S(\omega)$ is the distribution of the frequency deviations from the center of the resonance, i.e., it is the lineshape function. An ensemble of spins in the sample with equal atomic positions and equal spatial orientations (polar coordinates $\{\theta, \varphi\}$ of the $z$ axis of the tensor main frame) contributes to the spectrum at the two positions

$$\omega_{\mathrm{aniso}} = \pm \frac{D}{2} \left[ 3\cos^2\theta - 1 + \eta \sin^2\theta \cos 2\varphi \right] \tag{22}$$

The whole line shape function is the sum over all individual lines which are here assumed to be $\delta$ functions:

$$S(\omega) = \int\limits_0^{2\pi} \mathrm{d}\varphi \int\limits_0^{\pi} \mathrm{d}\theta \, \sin\theta \, \delta \left( \omega - \omega_{\mathrm{aniso}}(\theta, \varphi) \right) \tag{23}$$

Inserting eqns. (23) and (22) into eqn. (21) gives

$$M_2 = \frac{D_0^2}{5} \left( 1 + \frac{\eta^2}{3} \right) \tag{24}$$

Now we can use the effective anisotropies and asymmetry parameters of the four models discuss above and obtain

$$M_2^{(\mathrm{CH})} = \frac{D_0^2}{5} ; \qquad M_2^{(\mathrm{CH\text{-}j})} = \frac{7}{80} D_0^2 ; \qquad M_2^{(\mathrm{CH2})} = \frac{2}{5} D_0^2 ; \qquad M_2^{(\mathrm{CH3})} = \frac{1}{15} D_0^2 ; \tag{25}$$

The two last expressions were obtained as averages between the corresponding expressions from both sum and difference tensors mentioned above. They are equal to the expressions for the second moments of multispin dipolar interactions; in the case of $CH_3$ the fast-rotation of this group was considered.

## 2.11 Parameter fit to experimental data

From the model functions and the exponential-damping approximation, the following expressions can be obtained which are immediately applicable for the data fit in the four respective cases mentioned above ($r := \tau_{\mathrm{c}}^{-1}$):

$$F_1^{\mathrm{IS}} \left( D_{\mathrm{res}}, \omega_{\mathrm{r}}, r, t \right) = \exp\left\{ -\frac{3}{5} \frac{3\pi D_{\mathrm{res}}^2}{\omega_{\mathrm{r}}^3} \frac{r}{T_{\mathrm{r}}} \frac{t}{T_{\mathrm{r}}} \right\} \sum_{m=0}^{m_{\mathrm{max}}} C_m^{\mathrm{IS}} \left( \frac{D_{\mathrm{res}}}{\omega_{\mathrm{r}}} \right) \cdot \cos 2\pi m \frac{t}{T_{\mathrm{r}}} \tag{26}$$

$$F_2^{\mathrm{IS\text{-}j}} \left( D_{\mathrm{res}}, \omega_{\mathrm{r}}, r, t \right) = \exp\left\{ -\frac{21}{80} \frac{3\pi D_{\mathrm{res}}^2}{\omega_{\mathrm{r}}^3} \frac{r}{T_{\mathrm{r}}} \frac{t}{T_{\mathrm{r}}} \right\} \sum_{m=0}^{m_{\mathrm{max}}} C_m^{\mathrm{IS\text{-}j}} \left( \frac{D_{\mathrm{res}}}{\omega_{\mathrm{r}}} \right) \cdot \cos 2\pi m \frac{t}{T_{\mathrm{r}}} \tag{27}$$

$$F_3^{\mathrm{IS2}} \left( D_{\mathrm{res}}, \omega_{\mathrm{r}}, r, t \right) = \exp\left\{ -\frac{6}{5} \frac{3\pi D_{\mathrm{res}}^2}{\omega_{\mathrm{r}}^3} \frac{r}{T_{\mathrm{r}}} \frac{t}{T_{\mathrm{r}}} \right\} \sum_{m=0}^{m_{\mathrm{max}}} C_m^{\mathrm{IS2}} \left( \frac{D_{\mathrm{res}}}{\omega_{\mathrm{r}}} \right) \cdot \cos 2\pi m \frac{t}{T_{\mathrm{r}}} \tag{28}$$



$$F_4^{\mathrm{CH3}}\left(D_{\mathrm{res}}, \omega_{\mathrm{r}}, r, t\right) = \exp\left\{-\frac{1}{5}\frac{3\pi D_{\mathrm{res}}^2 r}{\omega_{\mathrm{r}}^3}\frac{t}{T_{\mathrm{r}}}\right\} \sum_{m=0}^{m_{\max}} C_m^{\mathrm{CH3}}\left(\frac{D_{\mathrm{res}}}{\omega_{\mathrm{r}}}\right) \cdot \cos 2\pi m \frac{t}{T_{\mathrm{r}}} \tag{29}$$

For each sample, data were recorded at different temperatures and different spinning speeds $f_n$. For each spinning speed, 8 data points were recorded along the indirect dimension $t_1$ at the times $0$, $T_{\mathrm{r}}/7$, $2T_{\mathrm{r}}/7$, .., $T_{\mathrm{r}}$. The fitting procedure consisted of minimizing the summed square deviation $\chi^2$ between data and model function

$$\chi^2_{\min} := \min_{r > 0,\, D_{\mathrm{res}} > 0} \chi^2 \qquad \text{with} \qquad \chi^2 := \sum_n \sum_{p=1}^{8}\left[y_p^{(n)} - F\left(D_{\mathrm{res}}, 2\pi f_n, r, t_p\right)\right]^2 \tag{30}$$

For obtaining information about the accuracy of the resulting parameters, the region in the $D_{\mathrm{res}}, r$ plane was considered where $\chi^2\left(D_{\mathrm{res}}, r\right) < 2\,\chi^2_{\min}$.

## 3 Experimental

### 3.1 $^{13}$C solid-state NMR spectroscopy

One part of the $^{13}$C solid-state NMR experiments (spectroscopy, HETCOR) was performed on a Bruker Avance NMR spec-
trometer operating at 75 MHz and 300 MHz for $^{13}$C and protons, respectively, using a commercial 2.5 mm MAS NMR probe. Ramped $^1$H-$^{13}$C cross-polarization (Pines et al., 1973; Metz et al., 1994) and SPINAL $^1$H-decoupling (Fung et al., 2000) was applied. The temperature inside the rotor was calibrated using the $^{207}$Pb chemical shift of lead nitrate as a temperature reference (Ferguson and Haw, 1995). Adamantane served as an external shift standard.

DIPSHIFT experiments were performed on a Bruker Avance III spectrometer operating at 100 MHz and 400 MHz for $^{13}$C
and protons, respectively, with a commercial 4 mm MAS NMR probe. The spinning speed was varied as described in the results section. The $^1$H nutation frequency was 60.5 kHz which required an offset of 42.8 kHz for the Lee-Goldburg decoupling within these experiments. Between 3200 and 6400 scans per $t_1$ value were recorded with recycle delays of 2 s (25°C) and 3 s (-15°C), resp.

### 3.2 Synthesis of the linker and of PIZOF-10

The PIZOFs were obtained through the standard process described for PEPEP-PIZOFs by Schaate et al. (2011). For the synthesis of PIZOF-10, ZrCl$_4$ (0.080 g, 0.34 mmol), HO$_2$C[PE-P(O(CH$_2$CH$_2$O)$_3$CH$_3$, O(CH$_2$CH$_2$O)$_3$CH$_3$)-EP]CO$_2$H (0.34 mmol) and benzoic acid (1.256 g, 10.29 mmol) as modulator were dissolved in 20 mL DMF. The solution was heated in a tightly capped 100 mL glass flask to 120 °C in an oven for 4 days. The precipitate was isolated from the mother liquor via centrifugation and washed by immersion in DMF for 30 min followed by centrifugation. The immersion and centrifugation steps
were repeated with ethanol. The product was dried at reduced pressure and the obtained dry powder was submitted to Soxhlet extraction with ethanol for 24 h. The mother liquor was again filled into a glass flask and heated to 120 °C. An additional





fraction of product was obtained within 24 hours, which was isolated and washed as described for the first crop. All fractions were merged.

## 4    Results

### 4.1    Spectral assignment of the $^{13}$C spectra

The solid-state $^{13}$C NMR spectra of PIZOF-2, PIZOF-10, and PIZOF-11 are well resolved. This allowed us to obtain almost complete $^{13}$C NMR signal assignment for these three samples (see Fig. 5, Figures S1 and S2 (Supplementing Information) and Table 1). Rough assigment of the solid-state $^{13}$C lines could be obtained already by a comparison to the liquid-state $^{13}$C NMR spectra of the dicarboxylic acid (used for the synthesis of PIZOF-10) in DMSO-$d_6$. We observed very similar chemical shifts

for most of the detected signals. Significant differences of the order of ca. 5 ppm occur for the carboxylate group (10) and the aromatic carbon (8) directly bound to (10). This is probably caused by the proximity to the metal atom. The signal assignment is further supported on the interpretation of liquid-state DEPT 135 as well as solid-state APT spectra (Fig. 5). Both methods are able to distinguish differently protonated carbons: In DEPT 135 spectra, the signals of CH and CH$_3$ appear positive, whereas CH$_2$ groups cause negative resonances. Quaternary carbons do not give any signal in DEPT 135. In analogy to the routine

liquid-state APT method, for a certain internal delay, CH and CH$_3$ groups produce negative signals while the resonances of quaternary carbons and CH$_2$ groups are positive. Distinction between quaternary carbons and CH$_2$ is also possible based on the significantly higher intensity of quaternary carbon signals compared to the signals of CH$_2$. The signal assignments summarized in Table 1 are further corroborated by $^{1}$H-$^{13}$C HETCOR and $^{1}$H-$^{13}$C HMQC spectra, see Fig. 6.

Comparing the solid-state $^{13}$C spectra of the different PIZOFs we found the most relevant differences between the spectra

of the different PIZOFs in the CH$_2$ region, showing the different kinds of sidechains, and in the aromatic region between 111 and 118 ppm. Whereas the PIZOF-2 spectrum simply contains the OCH$_3$ resonance, no complete resolution is found for the different aliphatic carbon positions in the chains except PIZOF-11 at room temperature. Further, the resonance of the CH of the central benzene ring overlaps with the line of $\underline{C}^{ar}C{\equiv}C$. They could be separated for PIZOF-10 and PIZOF-11, see below. For PIZOF-2, however, a complete overlap of both resonances occurs which does not permit a DIPSHIFT analysis of the

central-ring CH at all.

### 4.2    Indication for anisotropic motions of the sidechains

Figure 7 shows the directly excited $^{13}$C MAS NMR spectra as well as $^{13}$C CP MAS NMR spectra measured with 4 ms contact time for the three samples under study. If the long side chains in PIZOF-10 and PIZOF-11 were liquid-like and would move isotropically, the $^{1}$H-$^{13}$C CP efficiency should be very low or even zero as compared to the $^{13}$C signals of the other framework

carbons. The existence of rather strong aliphatic lines implies interpretations either as a consequence of slow motion of the chains, or as indeed fast but spatially restricted reorientation. The latter leads, even for very short correlation times, to a non-

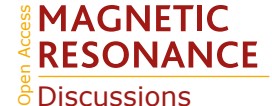

| Signal | $\delta$ [ppm] | Structural group |
|:------:|:--------------:|:----------------:|
| 1 | 58 | $OCH_3$ (1) |
| 2 | 71 | $OCH_2CH_2O$ (2) |
| 3 | 89.5 / 96.5 | $C{\equiv}C$ (3) |
| 4 | 115 | $\underline{C}^{ar}C{\equiv}C$ (ortho to O) (4) |
| 5 | 117 | $C^{ar}H$ (ortho to O) (5) |
| 6 | 127 | $C^{ar}H$ (para to $CO_2$) (6) |
| 7 | 131 | $C^{ar}H$ (7) |
| 8 | 134 | $\underline{C}^{ar}CO_2H$ (8) |
| 9 | 155 | $C^{ar}O$ (9) |
| 10 | 171 | $CO_2H$(10) |

**Table 1.** Assignment of the solid-state $^{13}C$ NMR signals of PIZOF-10

zero residual dipolar coupling, see above, which enables cross polarization even in the case of a fast motion. For distinguishing between these cases the DIPSHIFT method was used.

### 4.3 Line decompositions

As explained above, we focus on the signals of protonated carbons for the DIPSHIFT data. This concerns the CH of the lateral rings (overlapping at about 130 ppm with the resonances of some quarternary C) and the CH of the central ring. In PIZOF-10 and -11, the related lines overlap partially with the line of the quarternary C bound to the acetylenic C (in PIZOF-2 there is a complete overlap which could not resolved, see above). The sidechain resonances in PIZOF-10 and -11 feature some overlap; their resolution could deliver information about the order parameter gradient along the side chain.

It turns out that, despite the enhanced uncertainty because of decomposition, well-defined plots of intensity vs. $t_1/T_r$ were obtained. Concerning the aliphatic region of PIZOF-11, whereas at 30°C five $CH_2$ lines are well resolved, at -15 °C there is a strong overlap because of increased line broadening. Fig. 9 shows a decomposition in Lorentzian components. The lines could be assigned to the atomic positions by means of the predictions of ACD software (Advanced Chemistry Development Inc., 2017). Most lines at -15 °C can be assigned immediately to corresponding lines of the 30 °C spectrum with one exception:

The central resonance ($\delta$ position) seems to split at lower temperature into two overlapping components. The position of one of them corresponds to $\delta$ in the 30 °C spectrum, the other in a distance of about -1.2 ppm is denoted here as $\delta_1$. It is possible that different conformations (for example trans and gauche) along the chains are frozen-in at -15 °C whereas faster motions at 30 °C average them, leading to an average chemical shift.

In some contrast, the signal assignment for PIZOF-10 is not fully clear. From a comparison with a prediction of isotropic

chemical shifts obtained by the ACD software, we expect the resonances of carbons 3, 4, 5, 6 within a region of 0.7 ppm, carbon 2 to be shifted by about 1.2 ppm with respect to the center of the former group of resonances, and carbon 7 to be shifted by about -0.6 ppm. Taking into account that the solid-state resonances might deviate non-systematically from the solution-based





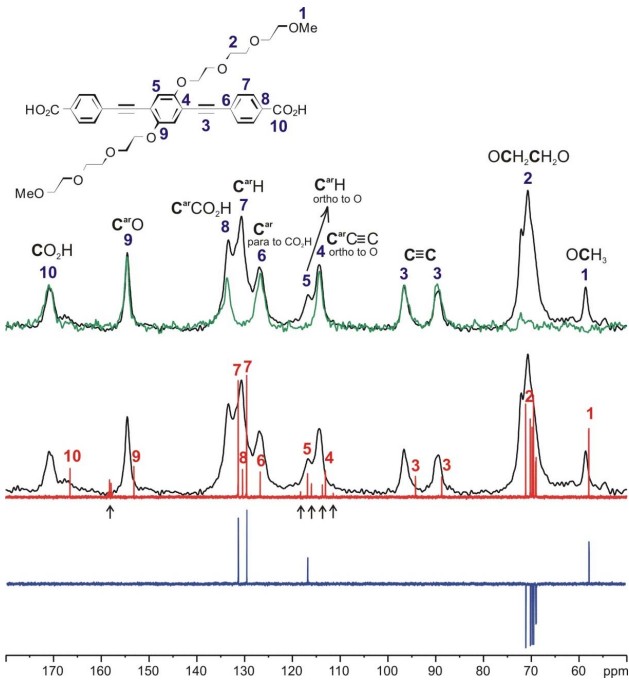

**Figure 5.** Signal assignment for the $^{13}$C NMR spectrum of the linker of PIZOF-10 based on the interpretation of $^{13}$C CP MAS NMR spectrum (black), $^{13}$C solid-state APT NMR spectrum (green), $^{1}$H broadband proton-decoupled liquid-state (DMSO-d6), $^{13}$C NMR spectrum (red), and liquid-state (DMSO-d6) DEPT 135 $^{13}$C NMR spectrum (blue). The signals marked with ↑ are assigned to trifluoroacetic acid, which was used in the last step of the synthesis of the linkers.

shifts, we could attain a coarse assignment of the -15° lines to the following carbon positions: A to 2, B to 3, 4, 5 and 6, C to position 7. For the five components of the 25°C spectrum, the assignment however will remain unknown at this stage. Typical

solid-state NMR phenomena like conformational differences (e.g. trans/gauche) could generate additional shifts. A possible assignment will be given in the discussion section.

The separation of the ring-CH resonances from overlapping quarternary C resonances is always straightforward. Further lineshape decompositions can be found in the Supplementary Material.

### 4.4 Evaluation of the CP buildup curves

Comparisons of the signal intensities for PIZOF-10 and PIZOF-11 reveal similar CP efficiencies for most side-chain carbon signals as for the other $^{13}$C signals of protonated carbons in the framework – except for the chain ends, where the CP efficiency significantly drops (Figure 7).

In CP experiments, several parameters influence the rate and efficiency of polarization transfer, in particular (i) the longitudinal relaxation time of the protons in the rotating frame $T_{1\rho}$, (ii) the number of neighboring protons and their distances

from the $^{13}$C nuclei, and (iii) the mobility of the nuclei. To investigate the spin polarization transfer, CP buildup curves are

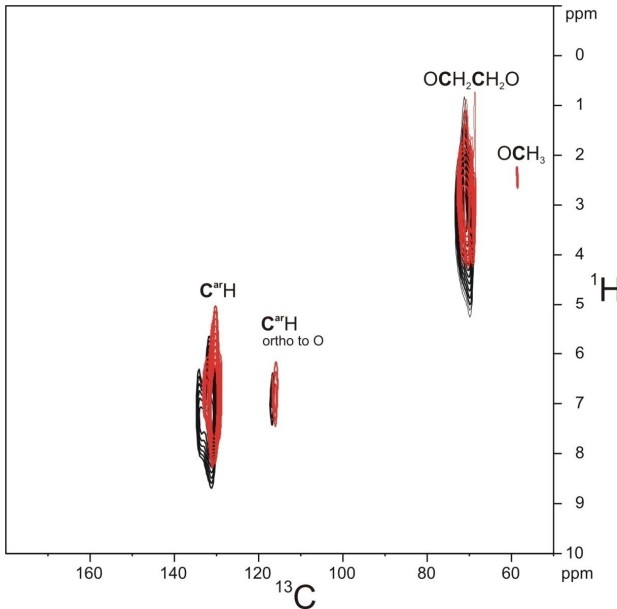

**Figure 6.** Comparison of the $^1$H-$^{13}$C HETCOR spectrum of PIZOF-10 measured at short contact time (0.7 ms, black) and the $^1$H-$^{13}$C HMQC spectrum (red).

| Signal | 1 | 2 | 3 | 4 | 5 | 6 | 7 | 8 | 9 | 10 |
|---|---|---|---|---|---|---|---|---|---|---|
| Group | OCH$_3$ | OCH$_2$ | C≡C | C$^{ar}$ | C$^{ar}$H | C$^{ar}$ | C$^{ar}$H | C$^{ar}$ | C$^{ar}$O | COOH |
| $T_{CH}$ in ms | 1.9 | 0.2 | 4.2 | 2.8 | 0.2 | 1.9 | 0.2 | 0.4 | 2.5 | 4.6 |

**Table 2.** Buildup time constants $T_{CH}$ values obtained for the $^{13}$C signals in PIZOF-10

measured (Mehring, 1983). $^{13}$C CP MAS spectra are recorded with different contact times (here between 0.35 and 10 ms) and the intensities of the individual signals are plotted versus the contact time. Protons directly bound to the carbon lead to fast and efficient CP buildup and hence to short buildup time constants $T_{CH}$, whereas carbons without directly bound protons, e.g., larger distances to protons show longer CP buildup times. Thermal motions result in a less efficient CP buildup and result in
longer CP buildup times $T_{CH}$ (Schulze et al., 1990). CP buildup time constants for PIZOF-10 are shown in Figure 8 and Table 2.

The fastest CP buildups are observed for aromatic CH groups (5, 7) and CH$_2$O groups (2). It is interesting to compare the CP buildup time constants for protonated carbon positions located at the framework (CH groups of the lateral rings 5, 7) with the time constants for the CH$_2$ groups of the side chains (2). If the rigidity of the framework and the sidechains were identical,
$T_{CH}$ of CH$_2$ would be shorter than for CH. In our experiments, the values for $T_{CH}$ of CH$_2$ and CH are similar. This indicates that the sidechains (2) are slightly more flexible than the framework (5, 7) but still relatively rigid. Because the signals of the different side-chain positions are not resolved (2), the time constant cannot be determined for the different CH$_2$ groups.





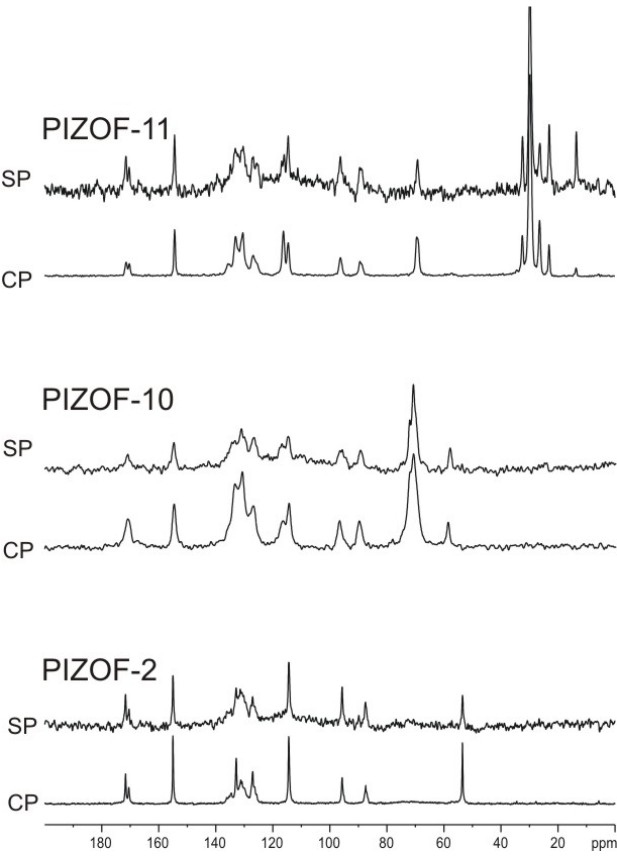

**Figure 7.** Comparison of directly excited $^{13}$C MAS NMR spectra and $^{13}$C CP MAS NMR spectra (4 ms contact time) of PIZOF-2, PIZOF-10 and PIZOF-11.

The DIPSHIFT measurements however allow further interpretations. Theoretically, an immobile CH$_3$ group should have the shortest time $T_{\mathrm{CH}}$ compared to all other carbons in PIZOF-10. The terminal CH$_3$ at the end of the side chains (1), however,
exhibits a rather long $T_{\mathrm{CH}}$ value. This is likely due to the combined influence of methyl group rotations and increased mobility of the chain end. It can, therefore, be concluded that the side chains are far from showing a liquid-like behavior. Obviously, they are relatively ordered and fixed in space due to interactions among themselves and/or with the framework. This is in agreement with previous $^{13}$C CP MAS NMR studies, e.g., of $n$-alkyl silanes bound to silica surfaces (Sindorf and Maciel, 1983) showing that (i) significant CP transfer occurs for all carbon atoms and (ii) the CP efficiency decreases with increasing distance from
the surface, i.e., from the anchoring point of the chain.

### 4.5 Evaluation of the DIPSHIFT curves

DIPSHIFT data and model functions with fitted parameters $D_{\mathrm{res}}$ and $r_{\mathrm{c}}$ of the methyl carbon in PIZOF-2 are shown in Fig. 11. For an impression of the quality of the fit, the $\chi^2(D_{\mathrm{res}}, r_{\mathrm{c}})$ surface is shown in Fig. 12, the minimum position being marked by





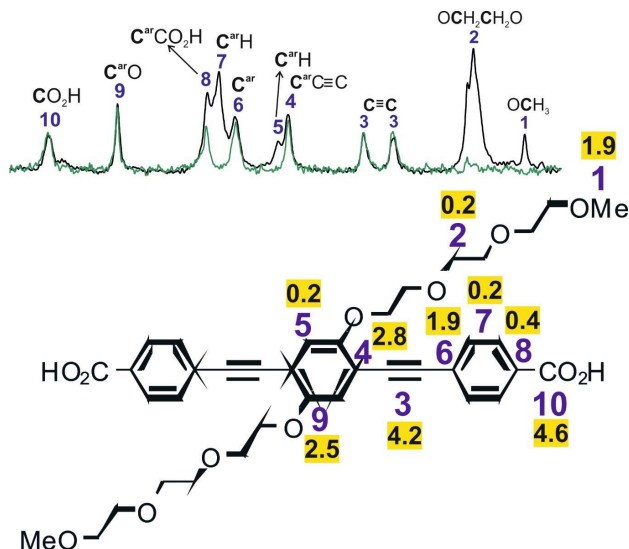

**Figure 8.** $^1$H-$^{13}$C CP buildup time constants (in units of ms) for selected $^{13}$C signals of PIZOF-10.

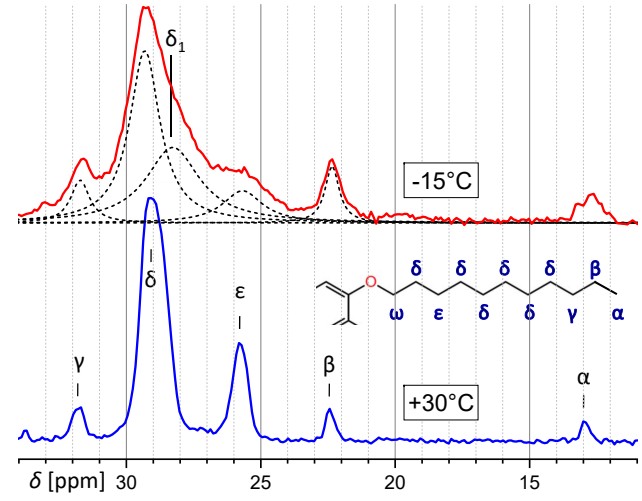

**Figure 9.** Comparison of the aliphatic regions of $^{13}$C spectra of PIZOF-11 at different temperatures. Whereas at 30°C different positions are well separated from another, it is not the case at -15°C. The decomposition into a number of Lorentzian lines is shown. One component of this decomposition ($\delta_1$) seems to have no counterpart at 30 °C. ($\omega$-CH$_2$ is at 69 ppm and not visible here.)

a cross. The closed line connects all points where $\chi^2$ is twice its minimum value. For a Gaussian distribution this would be the distance of one standard deviation from the minimum.

Numerical values for $D_{\text{res}}$ and $r$ are summarized in Tables 3, 4 and 5 for the respective samples. The last columns each contain the order parameters $S = D\text{res}/D_0$. C–H distances are between 1.09 and 1.10 Å which gives a coupling constant

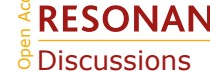

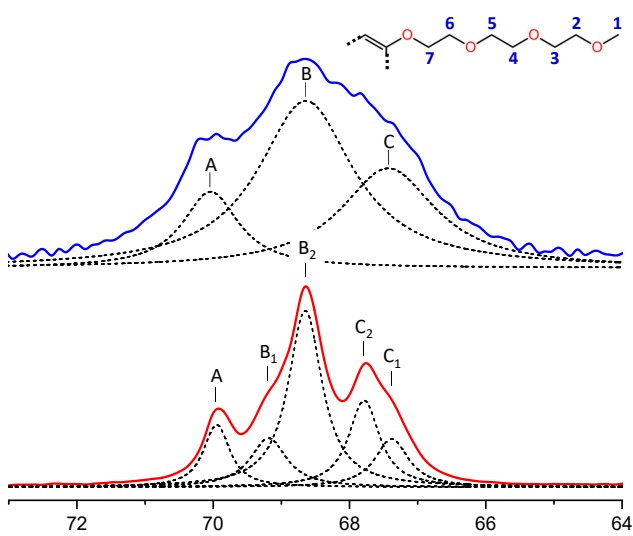

**Figure 10.** Comparison of the aliphatic regions of $^{13}$C spectra of PIZOF-10 at different temperatures. Whereas at 25°C five components could be resolved (bottom), at -15°C only three were detected (top). The numbering of the carbon positions serves to discussions in subsection "line decompositions".

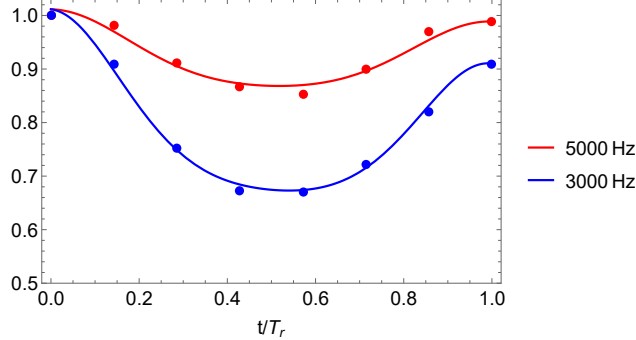

**Figure 11.** DIPSHIFT curves of OCH$_3$ of PIZOF-2 at 30°C for different spinning frequencies. The solid lines show model function 4 with values for the parameters $D_{\mathrm{res}}$ and $r_{\mathrm{c}}$ which give the best fit to the experimental data for both spinning rates.

reduced by the Lee-Goldburg scaling factor $D_{\mathrm{LG}} = D_0/\sqrt{3} = (13.28 \pm 0.18)$ kHz. The resulting uncertainty of about 1.4 % is smaller than the experimental uncertainties and therefore influences the uncertainty of the order parameter only weakly.

## 5  Discussion


The evolution of the $^{13}$C magnetization during a $^1$H-$^{13}$C cross-polarization experiment is characterized for short times ($\sim D_{\mathrm{res}}^{-1}$) by powder-averaged oscillations determined by the (residual) dipolar coupling. For longer times, however, spin diffusion within



| Carbon position | Temp. | fit model | $D_{\text{res}}$[kHz] | $r_{\text{c}}$[ms$^{-1}$] | $\chi^2_{\text{min}}$ | $S$ |
|---|---|---|---|---|---|---|
| CH$_3$ | +30°C | 4 | 5.8 ±0.5 | 0.4 ±0.4 | 0.00395 | 0.436 ± 0.028 |
| | -15°C | 4 | 6.47±0.37 | 0.68± 0.15 | 0.00147 | 0.486 ± 0.038 |
| CH | -15°C | 1 | 8.3 ±1.0 | 0.4 $^{+0.5}_{-0.3}$ | 0.0516 | 0.624 ± 0.075 |
| (terminal ring) | -15°C | 2 | 12.6 ±1.5 | . | 0.0528 | 0.947 ± 0.113 |

**Table 3.** Residual dipolar couplings, damping constants, minimum mean-square deviations and order parameters for methyl carbons and for protonated sidering carbons in PIZOF-2.

| Carbon position | Temp. | fit model | $D_{\text{res}}$/kHz | $r_{\text{c}}$ [ms$^{-1}$] | $\chi^2_{\text{min}}$ | $S$ |
|---|---|---|---|---|---|---|
| CH$_2$ A | -15°C | 3 | 2.9 ±0.5 | 3.3 $^{+12}_{-3.3}$ | 0.0225 | 0.218 ± 0.038 |
| | 25°C | 3 | 2.7 ±0.6 | 1.3 $^{+2.6}_{-1.3}$ | 0.00529 | 0.201 ± 0.045 |
| CH$_2$ B$_1$ | 25°C | 3 | 3.4 ±0.4 | 3. $^{+5.6}_{-3}$ | 0.00530 | 0.256 ± 0.030 |
| CH$_2$ B | -15°C | 3 | 3.11±0.25 | 0.8 $^{+1}_{-0.7}$ | 0.00767 | 0.234 ± 0.023 |
| CH$_2$ B$_2$ | 25°C | 3 | 2.08±0.35 | 0.9 $^{+6}_{-0.9}$ | 0.0296 | 0.156 ± 0.038 |
| CH$_2$ C$_2$ | 25°C | 3 | 4.64±0.3 | 1.4 $^{+1.6}_{-1.2}$ | 0.0052 | 0.349 ± 0.023 |
| CH$_2$ C | -15°C | 3 | 8.2 ±0.7 | 1.1 $^{+1.2}_{-0.8}$ | 0.0232 | 0.614 ± 0.053 |
| CH$_2$ C$_1$ | 25°C | 3 | 6.4 ±0.5 | 1.9 $^{+2}_{-1.4}$ | 0.0135 | 0.478 ± 0.038 |
| Middle ring | -15°C | 1 | 10.9 ±1.0 | 0.8 $^{+1.0}_{-0.6}$ | 0.050 | 0.823 ± 0.075 |
| | | 2 | 16.6 ±1.5 | ". " | 0.051 | 1.245 ± 0.113 |
| | 25°C | 1 | 10.7 ±0.9 | 0.4 $^{+0.8}_{-0.4}$ | 0.0492 | 0.805 ± 0.068 |
| | | 2 | 15.6 ±2.5 | ". " | 0.169 | 1.173 ± 0.188 |
| Terminal ring | -15°C | 1 | 10.2 ±0.9 | 0.5 $^{+0.8}_{-0.5}$ | 0.0283 | 0.767 ± 0.068 |
| | 25°C | 1 | 8.5 ±0.9 | 0.21$^{+0.5}_{-0.2}$ | 0.0637 | 0.639 ± 0.068 |
| | | 2 | 12.8 ±1.5 | ". " | 0.0649 | 0.962 ± 0.113 |

**Table 4.** Residual dipolar couplings, damping constants, minimum mean-square deviations and order parameters for some carbon positions and for components of methylene resonances in PIZOF-10. Again, for the ring CH, two different fit models were applied, see text.



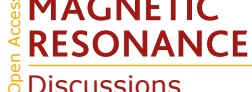

| Line assignment | Temp. | fit model | $D_{\mathrm{res}}$/kHz | $r$ [ms$^{-1}$] | $\chi^2_{\min}$ | $S$ |
|---|---|---|---|---|---|---|
| $\beta$-CH$_2$ | -15°C | 3 | 1.5 $^{+0.3}_{-0.5}$ | 0.5 $^{+4}_{-0.5}$ | 0.0389 | 0.11$^{+0.024}_{-0.04}$ |
| | 25°C | 3 | 1.0 $\pm$0.5 | 6.4 $^{+50}_{-6.0}$ | 0.0128 | 0.075 $^{+0.030}_{-0.045}$ |
| | 30°C | 3 | 1.2 $^{+0.3}_{-0.4}$ | 0.3 $^{+3.5}_{-0.3}$ | 0.0238 | 0.09 $^{+0.02}_{-0.03}$ |
| $\gamma$-CH$_2$ | -15°C | 3 | 1.88$\pm$ 0.5 | 0.50 $^{+3.3}_{-0.5}$ | 0.0630 | 0.14 $\pm$ 0.04 |
| | 25°C | 3 | 1.28$\pm$ 0.3 | 3.6 $^{+6}_{-3.6}$ | 0.0097 | 0.10 $\pm$ 0.03 |
| | 30°C | 3 | 1.6 $\pm$ 0.4 | 0.3 $^{+2}_{-0.3}$ | 0.0324 | 0.12$^{+0.02}_{-0.03}$ |
| $\delta$-CH$_2$ | -15°C | 3 | 2.5 $\pm$ 0.4 | 0.5 $^{+1.1}_{-0.5}$ | 0.0347 | 0.19 $\pm$ 0.03 |
| | -15°C | 3 | 5.0 $\pm$ 0.7 | 0.23 $^{+0.2}_{-0.1}$ | 0.0367 | 0.37 $\pm$ 0.06 |
| | 25°C | 3 | 1.94$\pm$ 0.16 | 1.4 $^{+0.8}_{-0.5}$ | 0.0048 | 0.15 $\pm$ 0.01 |
| | 30°C | 3 | 2.33$\pm$ 0.25 | 0.36 $^{0.4}_{-0.25}$ | 0.0246 | 0.17 $\pm$ 0.02 |
| $\varepsilon$-CH$_2$ | -15°C | 3 | 3.5 $\pm$ 0.5 | 0.31 $^{+0.8}_{-0.3}$ | 0.0374 | 0.26 $\pm$ 0.034 |
| | 25°C | 3 | 2.51$\pm$ 0.11 | 0.48 $\pm$0.25 | 0.0030 | 0.19 $\pm$ 0.01 |
| | 30°C | 3 | 2.44$\pm$ 0.2 | 0.34 $^{+0.5}_{-0.3}$ | 0.0129 | 0.18 $\pm$ 0.015 |
| $\omega$-CH$_2$ | -15°C | 3 | 7.1 $\pm$0.9 | 0.8 $^{+0.9}_{-0.5}$ | 0.1040 | 0.53 $\pm$ 0.068 |
| | 25°C | 3 | 5.5 $\pm$ 0.3 | 0.40 $\pm$0.16 | 0.0150 | 0.41 $\pm$ 0.02 |
| | 30°C | 3 | 5.55$\pm$ 0.45 | 0.22 $^{+0.13}_{-0.09}$ | 0.0247 | 0.42 $\pm$ 0.03 |
| Middle ring | -15°C | 1 | 9.9 $^{+1.1}_{-0.9}$ | 0.13 $^{+0.9}_{-0.13}$ | 0.1250 | 0.74$^{+0.090}_{-0.07}$ |
| | | 2 | 14.4 $\pm$3.1 | ." " | 0.1610 | 1.1 $\pm$ 0.02 |
| | 25°C | 1 | 10.0 $\pm$ 0.4 | 0.34 $\pm$0.3 | 0.0053 | 0.76 $\pm$ 0.03 |
| | | 2 | 15.2 $\pm$ 0.6 | ." " | 0.0055 | 1.14 $\pm$ 0.05 |
| | 30°C | 1 | 9.1 $\pm$ 1.1 | 0.16 $^{+0.64}_{-0.16}$ | 0.0567 | 0.68 $\pm$ 0.08 |
| | | 2 | 13.7 $\pm$ 1.8 | ." " | 0.0580 | 1.03 $\pm$ 0.14 |
| Terminal ring | -15°C | 1 | 9.9 $\pm$1.9 | 0.13 $^{+1.9}_{-0.13}$ | 0.1250 | 0.75 $\pm$ 0.14 |
| | 25°C | 1 | 9.4 $^{+1.25}_{-1.1}$ | 0.066$^{+0.3}_{-0.066}$ | 0.1530 | 0.71 $\pm$ 0.09 |
| | 30°C | 1 | 10.8 $\pm$ 1.3 | 0.27 $^{+0.55}_{-0.27}$ | 0.0729 | 0.80 $\pm$ 0.10 |

**Table 5.** Residual dipolar couplings, damping constants, minimum mean-square deviations and order parameters for some carbon positions in PIZOF-11. Again, for the ring CH, two different fit models were applied, see text.





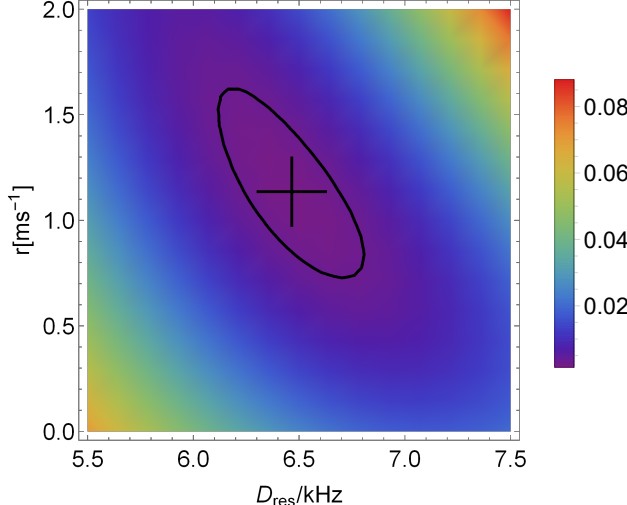

**Figure 12.** Mean square deviation $\chi^2$ between model function 4 and experimental DIPSHIFT data of CH$_3$ of PIZOF-2 vs. varied parameters $D_{\mathrm{res}}$ and damping constant $r_{\mathrm{c}}$. The cross marks the position of minimum $\chi^2$; the closed line marks the region where $\chi^2(D_{\mathrm{res}}, r_{\mathrm{c}}) = 2\chi^2_{\mathrm{min}}$.

the proton system will be more and more important; it provides a re-polarization of those protons which are close to $^{13}$C and have been depolarized during CP. Both phenomena are expected to be slowed down by thermal motion.

The $T_{\mathrm{CH}}$ values corresponding to the long-time buildup indicate an increased mobility towards the end of the side chains. With regards to the related thermal motion we can distinguish two cases: (i) Slow or intermediate nearly isotropic motion, which would cause a loss of CP efficiency upon speeding up; (ii) Fast anisotropic motion which does not hamper CP buildup even in the fast limit because of the residual dipolar coupling. The latter enables both oscillatory behaviour in the CH systems at short times and spin-diffusion in the proton system at long times. The DIPSHIFT results provide a distinction between these

two options. The first case would lead to strongly damped rotational echoes in the FID, see above; a re-increase in the second half of the DIPSHIFT curve would not occur. The DIPSHIFT signals obtained for the PIZOF samples, however, show a clear recovery of the rotational echo reaching almost the initial FID intensity.

In most cases investigated here, the residual dipolar couplings are essentially smaller than the value which would occur for immobile structural groups. Consequently, thermal motion that effectively averages the dipolar interactions must be present.

This motion must be much faster than the strength of carbon-proton dipolar coupling, i.e. the correlation times must be shorter than $10^{-5}$ s (in the intermediate region in a range $10^{-5}$ s ... $10^{-3}$ s, the motion would lead to a strong damping of the rotational echoes). The rotational echo after one rotation period is slightly damped, but in no case to less than 70% of the initial signal. That means that fast motion reduces the apparent dipolar coupling and the existence of a rather slow motion with correlation times of $2 \times 10^{-3}$ s causing the observed weak damping of the rotational echoes for room temperature as well as for -15°C. This

slow motion leads to a further averaging of the dipolar interaction, but just at times which are much longer than the duration of the DIPSHIFT modulation period.





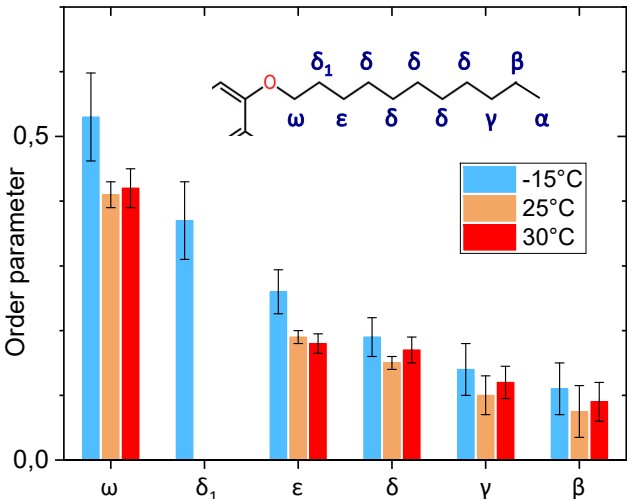

**Figure 13.** Graphical representation of the order parameter gradient along the side chains of PIZOF-11. The label "$\delta_1$" marks the position in the chain where the origin of this line is assumed; see text.

The sidechain order parameter could be estimated for different positions on the basis of our line decomposition, reminding that only for PIZOF-11 at 25°C, the aliphatic lines are well resolved. Besides the $\delta_1$ component occurring at -15°C only (see Fig. 9), all low-temperature components of the devonvolution could be assigned to the well-separated room temperature lines

and therefore also to carbon positions. A gradient of the order parameter along the chain could be detected, see Fig. 13. On this basis, we can give a possible explanation for the occurrence of the $\delta_1$ component. The value of its order parameter, which is placed between the $S$ values of $\varepsilon$ carbons and $\omega$ carbons, implies an assignment to the carbon being placed next to the $\omega$ carbon. At room temperature, its resonance is part of the signal group of the $\delta$ carbons. Perhaps because of conformational effects, it is shifted towards higher ppm values at -15°C.

The order parameters of the sidechains in PIZOF-10 are shown in Fig. 14. Following the liquid-state assignment (see "Line decompositions"), component A was assigned to pos. 2, component C was assigned to pos. 7. The order parameter of $\omega$-C is clearly larger than that of the remaining part of the side chain. This phenomenon occurring also in PIZOF-11 can be understood by the proximity of these carbons to the more rigid central rings, see below. The value of the order parameter of component $C_2$ strongly suggests an assignment of this component to position 6. The order parameter of component $B_1$ exceeds that of $B_2$

only weak yet beyond the limits of uncertainty. Hence, an assignment of this component to carbon position 5 is likely to some degree.

A comparison of the different side chains groups linked via O to the central rings shows that the order parameters of PIZOF-10 and of PIZOF-11 are equal within experimental uncertainty. However, in PIZOF-2, the bond order parameter of the methyl group is clearly smaller than that of the $\omega$-CH$_2$ of both other PIZOFs, reminding that we have considered the fast rotation

around the methyl symmetry axis in the fit. This indicates that there is a fast motion of the methyl axis itself which takes place in a range of angles which is of larger amplitude than in the case of the $\omega$ methylenes in PIZOF-10 and PIZOF-11.





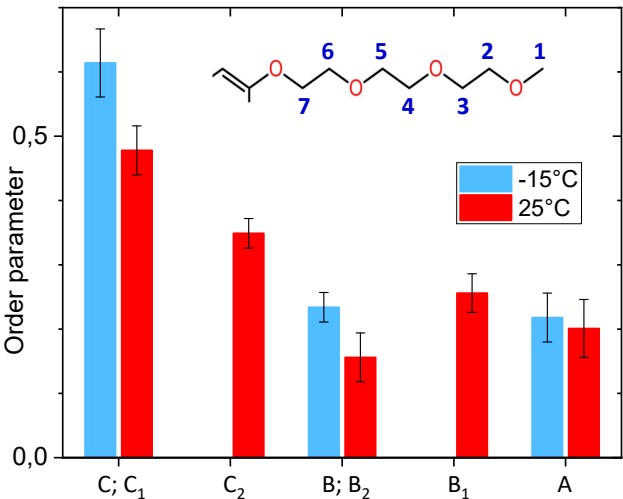

**Figure 14.** Graphical representation of the order parameters for the different components of the methylene line in PIZOF-10.

The order parameters of the central rings are graphically compared in Fig. 15. The PIZOF-2 spectrum did not allow for a separation of the corresponding line from a completely overlapped group of signals. Hence, we now deal with the two other samples only. For PIZOF-10, similar order parameters (0.74 and 0.68 for -15°C and 30°C, resp.) as for PIZOF-11 (0.82 and 0.80 for the same temperatures) were determined. We observe that the first ones are slightly smaller but not beyond the limits of experimental uncertainty. 180° flips of the rings are not expected because of the linked sidechains. Indeed, an attempt of evaluating the DIPSHIFT curves by model function 2 (for 120° jumps of the CH bonds) failed; it delivered unrealistically high values for the residual dipolar coupling.

With regards to the terminal rings, flips could also not be detected; the fit attempt led to $D_{res}$ values which would be larger than those expected for rigid molecular parts. This includes also the absence of slow flips, as this would lead to a larger $T_2$ damping of the rotational echo. An order parameter < 1 has thus to be explained by fast small-angle motions. For the central ring, $\pi$ flips have a very small likelihood because of the attached side chains.

## 6  Conclusions

Long aliphatic and ethyleneglycol sidechains of the PIZOF samples were shown to exhibit a gradient of anisotropy of motion. Not only the terminal methyl groups but also the methylene groups close to the $CH_3$ are subject to fewer restrictions than the inner $CH_2$ groups. The carbons closest to the linker backbone are subject to the strongest motional restrictions.

"Restriction" in our senses does not concern the speed of the motion but the degree of anisotropy. Assuming that this anisotropic motion would proceed as fluctuations of the $CH_2$ plane normals or $CH_3$ rotation axes within a cone with defined opening angle, this angle would be the larger, the smaller the order parameter is. This is reminiscent of the dynamics in





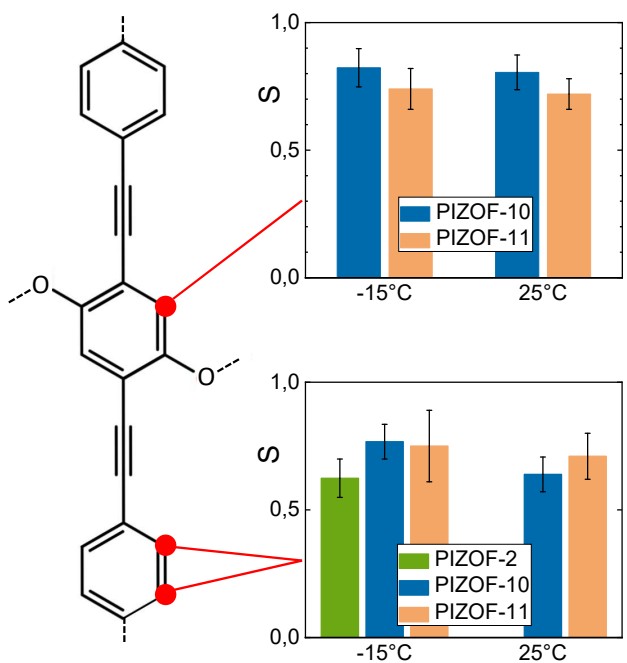

**Figure 15.** Graphical representation of the order parameters for the protonated carbons in the middle ring (top) and terminal rings (bottom).

nematic liquid crystals; overall we observe orientational restrictions to different degrees despite rather short correlation times. The difference to liquid crystals is found in the existence of long-range order of the chains of the latter.

# 7 Appendix

## 7.1 cos-Fourier coefficients

### 7.1.1 Ensemble of rigid $IS$ spin pairs

$$C_1^{\mathrm{IS}} = \frac{2}{15}\left(\frac{D_0}{\omega_\mathrm{r}}\right)^2 - \frac{17}{1260}\left(\frac{D_0}{\omega_\mathrm{r}}\right)^4 + \frac{131}{205\,920}\left(\frac{D_0}{\omega_\mathrm{r}}\right)^6 - \frac{12\,583}{705\,729\,024}\left(\frac{D_0}{\omega_\mathrm{r}}\right)^8$$


$$+ \frac{356\,981}{1\,072\,708\,116\,480}\left(\frac{D_0}{\omega_\mathrm{r}}\right)^{10} - \frac{25\,270\,187}{5\,693\,604\,618\,240\,000}\left(\frac{D_0}{\omega_\mathrm{r}}\right)^{12} \tag{31}$$

$$C_2^{\mathrm{IS}} = \frac{1}{60}\left(\frac{D_0}{\omega_\mathrm{r}}\right)^2 + \frac{1}{720}\left(\frac{D_0}{\omega_\mathrm{r}}\right)^4 - \frac{2983}{23\,063\,040}\left(\frac{D_0}{\omega_\mathrm{r}}\right)^6 + \frac{63\,863}{14\,114\,580\,480}\left(\frac{D_0}{\omega_\mathrm{r}}\right)^8$$

$$- \frac{268\,843}{2\,860\,554\,977\,280}\left(\frac{D_0}{\omega_\mathrm{r}}\right)^{10} + \frac{56\,352\,451}{42\,295\,348\,592\,640\,000}\left(\frac{D_0}{\omega_\mathrm{r}}\right)^{12} \tag{32}$$



$$C_3^{\text{IS}} = \frac{1}{1260} \left( \frac{D_0}{\omega_{\text{r}}} \right)^4 - \frac{397}{8\,648\,640} \left( \frac{D_0}{\omega_{\text{r}}} \right)^6 + \frac{353}{235\,243\,008} \left( \frac{D_0}{\omega_{\text{r}}} \right)^8$$
$$- \frac{17\,669}{536\,354\,058\,240} \left( \frac{D_0}{\omega_{\text{r}}} \right)^{10} + \frac{601\,529}{1\,174\,870\,794\,240\,000} \left( \frac{D_0}{\omega_{\text{r}}} \right)^{12} \tag{33}$$

### 7.2 Case 2: Spin pairs undergoing fast 2-site jumps

See equation (9):

$$C_0^{\text{IS-j}} = 1 - \frac{21}{320} \left( \frac{D_0}{\omega_{\text{r}}} \right)^2 + \frac{1589}{737\,280} \left( \frac{D_0}{\omega_{\text{r}}} \right)^4 - \frac{1\,887\,217}{47\,233\,105\,920} \left( \frac{D_0}{\omega_{\text{r}}} \right)^6 + \frac{11\,118\,847}{23\,492\,397\,367\,296} \left( \frac{D_0}{\omega_{\text{r}}} \right)^8$$
$$- \frac{757\,129\,037}{194\,773\,330\,899\,763\,200} \left( \frac{D_0}{\omega_{\text{r}}} \right)^{10} + \frac{9\,126\,896\,358\,637}{389\,583\,325\,014\,754\,590\,720\,000} \left( \frac{D_0}{\omega_{\text{r}}} \right)^{12} \tag{34}$$


$$C_1^{\text{IS-j}} = \frac{7}{120} \left( \frac{D_0}{\omega_{\text{r}}} \right)^2 - \frac{119}{46\,080} \left( \frac{D_0}{\omega_{\text{r}}} \right)^4 + \frac{321\,983}{5\,904\,138\,240} \left( \frac{D_0}{\omega_{\text{r}}} \right)^6 - \frac{4\,601\,521}{6\,607\,236\,759\,552} \left( \frac{D_0}{\omega_{\text{r}}} \right)^8$$
$$+ \frac{193\,149\,817}{32\,137\,599\,598\,460\,928} \left( \frac{D_0}{\omega_{\text{r}}} \right)^{10} - \frac{46\,589\,311\,577\,279}{1\,241\,796\,848\,484\,530\,257\,920\,000} \left( \frac{D_0}{\omega_{\text{r}}} \right)^{12} \tag{35}$$

$$C_2^{\text{IS-j}} = \frac{7}{960} \left( \frac{D_0}{\omega_{\text{r}}} \right)^2 + \frac{49}{184\,320} \left( \frac{D_0}{\omega_{\text{r}}} \right)^4 - \frac{257\,057}{18\,893\,242\,368} \left( \frac{D_0}{\omega_{\text{r}}} \right)^6 + \frac{32\,474\,753}{132\,144\,735\,191\,040} \left( \frac{D_0}{\omega_{\text{r}}} \right)^8$$
$$- \frac{1\,107\,477\,091}{428\,501\,327\,979\,479\,040} \left( \frac{D_0}{\omega_{\text{r}}} \right)^{10} + \frac{91\,276\,751\,182\,213}{4\,967\,187\,393\,938\,121\,031\,680\,000} \left( \frac{D_0}{\omega_{\text{r}}} \right)^{12} \tag{36}$$

$$C_3^{\text{IS-j}} = \frac{7}{46\,080} \left( \frac{D_0}{\omega_{\text{r}}} \right)^4 - \frac{14\,123}{7\,084\,965\,888} \left( \frac{D_0}{\omega_{\text{r}}} \right)^6 - \frac{76\,733}{11\,012\,061\,265\,920} \left( \frac{D_0}{\omega_{\text{r}}} \right)^8$$
$$+ \frac{30\,099\,811}{80\,343\,998\,996\,152\,320} \left( \frac{D_0}{\omega_{\text{r}}} \right)^{10} - \frac{587\,419\,421\,473}{137\,977\,427\,609\,392\,250\,880\,000} \left( \frac{D_0}{\omega_{\text{r}}} \right)^{12} \tag{37}$$

### 7.3 Case 3: Spin triples IS2

$$C_0^{\text{IS2}} = 1 - \frac{3}{10} \left( \frac{D_0}{\omega_{\text{r}}} \right)^2 + \frac{227}{4536} \left( \frac{D_0}{\omega_{\text{r}}} \right)^4 - \frac{49\,471}{9\,729\,720} \left( \frac{D_0}{\omega_{\text{r}}} \right)^6 + \frac{293\,281}{840\,647\,808} \left( \frac{D_0}{\omega_{\text{r}}} \right)^8$$
$$- \frac{286\,311\,167}{16\,662\,021\,667\,200} \left( \frac{D_0}{\omega_{\text{r}}} \right)^{10} + \frac{142\,610\,164\,787}{225\,135\,039\,577\,536\,000} \left( \frac{D_0}{\omega_{\text{r}}} \right)^{12} \tag{38}$$


$$C_1^{\text{IS2}} = \frac{4}{15} \left( \frac{D_0}{\omega_{\text{r}}} \right)^2 - \frac{34}{567} \left( \frac{D_0}{\omega_{\text{r}}} \right)^4 + \frac{2843}{405\,405} \left( \frac{D_0}{\omega_{\text{r}}} \right)^6 - \frac{17\,689}{33\,776\,028} \left( \frac{D_0}{\omega_{\text{r}}} \right)^8$$
$$+ \frac{2\,272\,579}{83\,310\,108\,336} \left( \frac{D_0}{\omega_{\text{r}}} \right)^{10} - \frac{258\,076\,147}{246\,858\,596\,028\,000} \left( \frac{D_0}{\omega_{\text{r}}} \right)^{12} \tag{39}$$



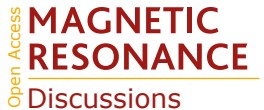

$$C_2^{\mathrm{IS2}} = \frac{1}{30}\left(\frac{D_0}{\omega_r}\right)^2 + \frac{1}{162}\left(\frac{D_0}{\omega_r}\right)^4 - \frac{2437}{1\,297\,296}\left(\frac{D_0}{\omega_r}\right)^6 + \frac{966\,239}{4\,728\,643\,920}\left(\frac{D_0}{\omega_r}\right)^8$$
$$- \frac{2\,084\,551}{158\,685\,920\,640}\left(\frac{D_0}{\omega_r}\right)^{10} + \frac{6\,874\,356\,527}{11\,938\,979\,371\,536\,000}\left(\frac{D_0}{\omega_r}\right)^{12} \tag{40}$$

$$C_3^{\mathrm{IS2}} = \frac{2}{567}\left(\frac{D_0}{\omega_r}\right)^4 - \frac{229}{1\,459\,458}\left(\frac{D_0}{\omega_r}\right)^6 - \frac{9379}{394\,053\,660}\left(\frac{D_0}{\omega_r}\right)^8$$
$$+ \frac{224\,699}{69\,425\,090\,280}\left(\frac{D_0}{\omega_r}\right)^{10} - \frac{19\,307\,643\,899}{98\,496\,579\,815\,172\,000}\left(\frac{D_0}{\omega_r}\right)^{12} \tag{41}$$

### 7.4 Case 4: Spin quartetts IS3

$$C_0^{\mathrm{IS3}} = 1 - \frac{1}{20}\left(\frac{D_0}{\omega_r}\right)^2 + \frac{227}{77\,760}\left(\frac{D_0}{\omega_r}\right)^4 - \frac{67\,283}{560\,431\,872}\left(\frac{D_0}{\omega_r}\right)^6 + \frac{12\,503\,873}{3\,919\,820\,636\,160}\left(\frac{D_0}{\omega_r}\right)^8$$


$$- \frac{122\,411\,059}{2\,116\,703\,143\,526\,400}\left(\frac{D_0}{\omega_r}\right)^{10} + \frac{4\,812\,476\,571\,577}{6\,357\,267\,735\,755\,489\,280\,000}\left(\frac{D_0}{\omega_r}\right)^{12} \tag{42}$$

$$C_1^{\mathrm{IS3}} = \frac{2}{45}\left(\frac{D_0}{\omega_r}\right)^2 - \frac{17}{4860}\left(\frac{D_0}{\omega_r}\right)^4 + \frac{7991}{50\,038\,560}\left(\frac{D_0}{\omega_r}\right)^6 - \frac{6\,882\,901}{1\,543\,429\,375\,488}\left(\frac{D_0}{\omega_r}\right)^8$$
$$+ \frac{13\,208\,297}{158\,752\,735\,764\,480}\left(\frac{D_0}{\omega_r}\right)^{10} - \frac{1\,119\,140\,771\,669}{1\,008\,604\,977\,307\,361\,280\,000}\left(\frac{D_0}{\omega_r}\right)^{12} \tag{43}$$

$$C_2^{\mathrm{IS3}} = \frac{1}{180}\left(\frac{D_0}{\omega_r}\right)^2 + \frac{7}{19\,440}\left(\frac{D_0}{\omega_r}\right)^4 - \frac{181\,963}{5\,604\,318\,720}\left(\frac{D_0}{\omega_r}\right)^6 + \frac{34\,933\,061}{30\,868\,587\,509\,760}\left(\frac{D_0}{\omega_r}\right)^8$$
$$- \frac{9\,947\,191}{423\,340\,628\,705\,280}\left(\frac{D_0}{\omega_r}\right)^{10} + \frac{2\,495\,680\,997\,437}{7\,492\,494\,117\,140\,398\,080\,000}\left(\frac{D_0}{\omega_r}\right)^{12} \tag{44}$$

$$C_3^{\mathrm{IS3}} = \frac{1}{4860}\left(\frac{D_0}{\omega_r}\right)^4 - \frac{24\,217}{2\,101\,619\,520}\left(\frac{D_0}{\omega_r}\right)^6 + \frac{193\,091}{514\,476\,458\,496}\left(\frac{D_0}{\omega_r}\right)^8$$
$$- \frac{653\,753}{79\,376\,367\,882\,240}\left(\frac{D_0}{\omega_r}\right)^{10} + \frac{26\,639\,914\,823}{208\,124\,836\,587\,233\,280\,000}\left(\frac{D_0}{\omega_r}\right)^{12} \tag{45}$$

## 8 Acknowledgements

We gratefully acknowledge MOF sample synthesis by Professor Peter Behrens and his co-workers Jann Lippke and Andreas
Schaate (Leibniz Universität Hannover, Institut für Anorganische Chemie). The linkers used for the MOF samples were syn-
thesized by Professor Adelheid Godt and Thomas Preuße (Universität Bielefeld, Fakultät für Chemie). We also thank Adelheid
Godt for carefully reading the manuscript and her valuable suggestions.



*Author contributions.* GH conducted theoretical considerations and data analysis, RK carried out the DIPSHIFT experiments and was involved in data analysis, SP performed the CPMAS and HETCOR experiments, KS and EB supervised the research, discussed the results. All authors contributed to paper writing and approved the final manuscript version.

*Data availability.* Experimental data are available upon request from the corresponding author.

*Competing interests.* Kay Saalwächter is a member of the editorial board of Magnetic Resonance.





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
