# Peer review of "Solid-state 13C-NMR spectroscopic determination of sidechain mobilities in zirconium-based metal-organic frameworks"

_Magnetic Resonance, 2023_

## Author Response (AR1)

Many thanks to the two referees! Thus, some inadequacies and typos could still be eliminated.

**Reply to RC1:**

RC1: l. 111: the authors must precise what are the relevant times for $^1H \rightarrow ^{13}C$ CPMAS and DIPSHIFT experiments.

Reply: We add the following sentences: For DIPSHIFT experiments, this is the time between the end of excitation and the beginning of data acquisition. Thus, the averaging must already be complete at the first time step in the indirect dimension which is one order of magnitude smaller than the rotation period. Completion of the averaging at $10^{-5}$ s means that the correlation time should be less than or at most equal to $10^{-6}$ s. In the case of cross-polarization, this relevant time is of the order of the reciprocal of the dipolar coupling constant, i.e. also about $10^{-5}$ s.

RC1: l. 174: the example chosen by the author lacks clarity since according to Eq. 9, $C_m^{IS-j} = C_m$. The anisotropy is only reduced by a scaling factor.

Reply: However, the asymmetry parameter is changed! To express this more clearly, the following sentence is added: This means that the asymmetry is reduced, but at the same time the asymmetry parameter increases from 0 (case of $C_m^{IS}$) to 3/5 ($C_m^{IS-j}$). Thus, the tensor principal values of one case can no longer be represented as common multiples of the principal values of the other.

RC1: l. 205: the authors must explain how 0.5 value was obtained since $\sqrt{5}/(2\pi) \approx 0.36$.

Reply: The value 0.5 will be replaced by 0.36.

RC1: l. 212: should the symbol "<<" be replaced by ">>"?

Reply: Misunderstanding of the reviewer? The correct symbol „$\gg$" is already present in the text.

RC1: l. 261: several experimental parameters of CP-HETCOR experiments are missing. The authors must indicate the employed number of scans, recycle delay and rf fields.

Reply: A sentence containing these parameters has been inserted.

RC1: l. 267: several experimental parameters of DIPSHIFT experiments are missing. The authors must indicate the rf fields.

Reply: Parameters were added.

RC1: Section 3.1: the authors must provide the experimental parameters, including recycle delays, number of scans and rf fields, for the solid-state $^1H$-$^{13}C$ APT, $^1H$-$^{13}C$ through-bond HMQC and $^{13}C$ direct excitation.

Reply: The missing parameters have been inserted.

RC1: l. 368: given the rf field inhomogeneity, there is an uncertainty on the angle between static and effective rf field and hence, on the scaling factor $\sqrt{3}$. The uncertainty on this scaling factor must be discussed with its influence on the uncertainty of the order parameter.

Reply: This influence is quite small. Therefore, a paragraph is inserted to explain this briefly and to point to a more detailed explanation in the SI. This detailed estimation of the measurement uncertainty will be included in the SI.

RC1: l. 389: the authors must explain how the correlation time value of $2 \times 10^{-3}$ s was determined.

Reply: The part "… with correlation times of $2 \times 10^{-3}$ …" s has been deleted.

RC1: l. 408-409: this sentence is not valid for PIZOF-11 at 30 °C. The authors must discuss that point.

Reply: Indeed, this sentence is valid only for – 15 °C. This temperature value is added to the sentence.

Additional comments:

RC1: l. 40: the reference Fung et al, 2000 is not related to CPMAS.

Reply: This citation is considered CP as a method. For this purpose, it is sufficient to cite the paper by Pines et al. The other two citations are therefore omitted in the revised version.

RC1: l. 48: the pulse sequence of DIPSHIFT must be displayed in the SI.

Reply: This pulse scheme has been added to the SI.

RC1: l. 53: the term "dipolar on-resonance FID" lacks clarity. The authors must precise that the $^{13}C$ magnetization solely evolves under $^{1}H$-$^{13}C$ dipolar couplings.

Reply: The last part of this sentence following „corresponding" was replaced by „"corresponding to an FID which solely evolves under MAS-modulated $^{1}H – ^{13}C$ dipolar couplings between subsequent rotational echoes."

RC1: l. 55: the term "mutual interaction" must be clarified.

Reply: It was changed to „interactions with each other".

RC1: l. 65: a reference must be cited after "Anderson-Weiss procedure".

Reply: We add the citation of the Andersson-Weiss paper at this text position (already cited below).

RC1: l. 102: the symbol ":=" must be defined.

Reply: Combinations of an equal sign with a colon are used in mathematics to represent a definition of one side by the other side. So this is a common notation. However, for sake of clarity, "where" in line 101 is replaced by "with the following definitions".

RC1: l. 111: the opening quotation marks are superscripted in English.

Reply: Corrected.

RC1: l. 159: "triple-bonded" adjective must be replaced by "acetylenic" since these carbon atoms are covalently bonded to two carbon atoms with four covalent bonds in the Lewis structure.

Reply: This replacement has been performed.

RC1: l. 167: the term "eigenvalue spectrum", which is not commonly used, must be defined or the sentence must be revised.

Reply: The sentence was shortened; it begins now with „All three eigenvalues of the total dipolar tensor…".

RC1: Caption of Figure 3 must be revised.

Reply: Revised caption of Fig. 3: The first sentence is replaced by: $^{1}H$ positions within a $CH_2$ group.

RC1: l. 190, Eq. 17: $M_2$ must be defined.

Reply: The first sentence in line 185 is changed into: „$K(\tau) = <\omega(t)\omega(t+\tau)>$ is the autocorrelation function of the dipolar frequency, related to the motion under consideration. For $\tau = 0$ this expression represents the second moment of the line shape, denoted by $M_2$, see below."

RC1: l. 196: this sentence lacks clarity and must be revised.

Reply: This sentence has been reworded to make it clearer.

RC1: l. 205: the author must refer to the equation number corresponding to "model function 1".

Reply: „(equation (5))" is added after „model function 1" in line 205.

RC1: l. 214, Eq. 18: $f_r$ symbol must be defined.

Reply: „and $f_r$ is the spinning frequency of the sample" was added in line 205 immediately before the semicolon.

RC1: l. 244, Eq. 26: $D_{res}$ symbol must be defined.

Reply: This symbol was defined already in line 169, hence it seems to be not required again here.

RC1: l. 315: this sentence must moved into the previous paragraph.

Reply: This sentence has been moved into the previous paragraph.

RC1: Figure 5: the structure of the molecule must be redrawn.

Reply: The quality of the molecule structure could be improved by enlarging it.

RC1: l. 337: the drop in CP efficiency for the chain ends is only visible for PIZOF-11 on Figure 7.

Reply: We insert "in Pizof –11" after "… for the chain ends" in this sentence.

RC1: l. 341: the authors must use the notation "$^1H\rightarrow^{13}C$ CP MAS" instead of "$^{13}C$ CP MAS".

Reply: To avoid any misunderstandings, we follow this advice.

RC1: Figure 10: to avoid confusion, the authors must use atom numbers in Figure 10, which differs from those used Figure 5.

Reply: Use of 2a, 2b, … , to be consistent with Fig. 5.

RC1: Figure 10: the axis label is missing. The authors must also display the temperatures on the figure.

Reply: Temperatures and axis label were added.

RC1: Caption of Table 3: the authors must precise that two different fit models were applied for ring CH.

Reply: A corresponding sentence was added to this caption.

RC1: l. 401: for the sake of consistency, the atom label ω-C must be replaced by 7.

Reply: „ω-C" was replaced by „the carbon at position 7".

RC1: Appendix: the numbering of the section is not consistent.

Reply: The appendix section was renamed to „cos-Fourier coefficients". The former subsubsection 7.1.1 is now subsection 7.1.

RC1: l. 444: 2 must be subscripted in the title of section 7.3; l. 449: 3 must be subscripted in the title of section 7.4.

Reply: Both numbers are subscribed.

RC1: For the sake of clarity, Figures in SI must be labeled as Figure Si with i = 1, 2, 3, etc.

Reply: This could be done only by redefining the TEX command which defines the caption format. I hope that the editor will accept this.

RC1: SI, Figures 2-3: the assignment of all peaks using atom numbering of Figure 5 must be provided.

Typos:

RC1:

- Caption of Figure 1. Add a colon after PIZOF's and a period at the end of the sentence.
- l. 37: $^{45}Sc$.
- l. 38: suppress space after "host/".

- l. 46: S must be italicized.
- l. 52: remove parentheses around 2019.
- l. 59: "was" instead of "could be".
- l. 147: the table number must be corrected.
- l. 149: the table caption is missing.
- Caption of Table 1: a period is missing.
- l. 316: a space must be inserted between "30" and "°C".
- ll. 319-322: hyphen must be replaced by minus symbol.
- l. 328: "C" is missing after "°".
- l. 367: "res" must be subscripted.
- Caption of Figure 11: a space must be inserted between "30" and "°C".
- l. 389: hyphen must be replaced by minus symbol.
- l. 393: a space must be inserted between "25" and "°C".
- SI, captions of Figures 1-3: the closing parenthesis must be removed at the end of the caption.
- SI, captions of Figures 4-24: periods are missing at the end of the figure captions.

Reply: All typos have been removed.

**Reply to RC2:**

RC2: The authors mention in several places the fast motion limit "... that motional averaging is completed at time which are relevant for the experiment". To me it is not so clear what the relevant time scale of the motion is. Can the authors elaborate on this question? Is it the time scale of MAS where averaging happens or the time scale of the dipolar coupling that dominates the time evolution? Does the rf-field amplitude in the DIPSHIFT experiment play an important role?

> Reply: We add the following sentences: For DIPSHIFT experiments, this is the times between the end of excitation and the beginning of data acquisition. Thus, the averaging must already be complete at the first time step in the indirect dimension which is one order of magnitude smaller than the rotation period, i.e. about 20 µs. In the case of cross-polarization, this relevant time is of the order of the reciprocal of the dipolar coupling constant, i.e. also about $10^{-5}$ s.

RC2: The manuscript has a lot of figures and, at least in the paper draft, the distance between figures and mentioning in the text is quite large which makes reading the paper cumbersome. I wonder whether some of the figures in the main text are really essential for reading the paper. For example, Fig. 12 could be moved to the SI without loss in readability. Of course, figure placement might be better in the final typeset version but still it might be better to reduce the total number or combine some of the figures.

> Reply: Fig. 12 has been moved to the SI.

RC2: I would recommend merging the Appendix into the SI. I feel it makes no sense to have both an Appendix and a SI.

> Reply: Chapter 7 contains information, especially equations, the knowledge of which is necessary for the understanding of the content. This chapter is renamed to „cos-Fourier coefficients" and given the heading of Section 7.1, see also Referee 1's comment in this regard.

Some minor details:

RC2: Figs. 13+14: It would be nice to indicate in Figs. 13 and 14 where the shown fragments are connected to the rigid part of the molecule by a "wavy" line.

> Reply: The images of the molecule fragments in this figures will be improved according to this comment.

RC2: Fig. 14: In Fig. 14 the tentative assignment of the different lines (A,B,C) to the numbered carbon atoms would be very helpful and informative. Of course this information is given in the text (but not the caption) but visualizing this in the figure would be helpful.

> Reply: This information will be inserted in the figure caption.

RC2: line 80: Why is fast anisotropic thermal motion only relevant for dipolar couplings and not also for CSA tensors? This is at least how I read the first point which might not be intended.

> Reply: The first item of the listing below is moved to immediately after equation (1) so that this statement includes also CSA tensors.

RC2: line 122: Why has \eta absolute value signs? \eta is, at least in the most commonly used convention of Mehring and Spiess, always between 0 and 1.

> Reply: The absolute value signs are omitted.

RC2: line 245: The cos functions should have parenthesis around the arguments.

> Reply: The cosine arguments in equations (26) to (29) get parantheses.

RC2: Fig. 12: The y axis label looks wrong.

> Reply: Probably you mean the contradiction to the figure caption where the damping variable is denoted by $r_c$. The latter is changed into $r$. (Corresponding to another comment of RC2, the figure was moved to SI.)

RC2: line 420: How large would be the scaling factor for ring flips? Given the large uncertainties of the determined order parameters, I wonder how far off these are from the experimental values. Of course, the averaged dipolar coupling would no longer be axially symmetric in a two-site jump model but this might be difficult to detect.

> Reply: $D_{res}$ was not obtained from the scaling of the curves compared to those of the rigid lattice, but from parameter fitting. The claim to have demonstrated a fast 2-site jump motion would only be possible if $D_{res} < D_{LG}$ had been found to be outside the error limits. To make this clearer, the following sentence was added: "Evidence of such rapid motion would only be given if $D_{res} < D_{LG}$ had been found to be outside the error limits." The beginning of the following sentence has been changed to "This concerns also slow flips, …".